# Longitudinal impact on rat cardiac tissue transcriptomic profiles due to acute intratracheal inhalation exposures to isoflurane

Sung-Hyun Park[1]*, Yuting Lu[2], Yongzhao Shao[2], Colette Prophete[1], Lori Horton[1], Maureen Sisco[1], Hyun-Wook Lee[1], Thomas Kluz[1], Hong Sun[1], Max Costa[1], Judith Zelikoff[1], Lung-Chi Chen[1], Mitchell D. Cohen[1]

1 Department of Environmental Medicine, New York University Grossman School of Medicine, New York, NY, United States of America, 2 Departments of Population Health & Environmental Medicine, New York University Grossman School of Medicine, New York, NY, United States of America

* SungHyun.Park@nyulangone.org

**Data Availability Statement:** All relevant data are within the manuscript and its Supporting Information files.

## Abstract

Isoflurane (ISO) is a widely used inhalation anesthetic in experiments with rodents and humans during surgery. Though ISO has not been reported to impart long-lasting side effects, it is unknown if ISO can influence gene regulation in certain tissues, including the heart. Such changes could have important implications for use of this anesthetic in patients susceptible to heart failure/other cardiac abnormalities. To test if ISO could alter gene regulation/expression in heart tissues, and if such changes were reversible, prolonged, or late onset with time, SHR (spontaneously hypertensive) rats were exposed by intratracheal inhalation to a 97.5% air/2.5% ISO mixture on two consecutive days (2 hr/d). Control rats breathed filtered air only. On Days 1, 30, 240, and 360 post-exposure, rat hearts were collected and total RNA was extracted from the left ventricle for global gene expression analysis. The data revealed differentially-expressed genes (DEG) in response to ISO (compared to naïve control) at all post-exposure timepoints. The data showed acute ISO exposures led to DEG associated with wounding, local immune function, inflammation, and circadian rhythm regulation at Days 1 and 30; these effects dissipated by Day 240. There were other significantly-increased DEG induced by ISO at Day 360; these included changes in expression of genes associated with cell signaling, differentiation, and migration, extracellular matrix organization, cell-substrate adhesion, heart development, and blood pressure regulation. Examination of consistent DEG at Days 240 and 360 indicated late onset DEG reflecting potential long-lasting effects from ISO; these included DEG associated with oxidative phosphorylation, ribosome, angiogenesis, mitochondrial translation elongation, and focal adhesion. Together, the data show acute repeated ISO exposures could impart variable effects on gene expression/regulation in the heart. While some alterations self-resolved, others appeared to be long-lasting or late onset. Whether such changes occur in all rat models or in humans remains to be investigated.

**Funding:** This work was supported by CDC/NIOSH Grant OH008280 and, in part, by NIEHS Center Grant ES00260.

**Competing interests:** The authors have declared that no competing interests exist.

## Introduction

Isoflurane (ISO) is a commonly used inhalation anesthetic in rodents for laboratory animal experiments [1] and on humans during some elective surgeries (though sevoflurane is increasingly being used in place of ISO) [2, 3]. Even though ISO is believed to usually maintain better cardiac function than other anesthetics (such as a combination of ketamine and xylazine), it still can give rise to various cardiovascular/cardiopulmonary side effects [4, 5]. Some of these have included respiratory depression, lowering of host blood pressure, and induction of an irregular heartbeat [6]. ISO is also known to induce changes in mean arterial pressure (MAP), heart rate (HR), and body temperature, as well as cause alterations in cardiac electrophysiologic function, arrhythmia, decreased cardiac contractility, and reduced cardiac force development [4, 7, 8]. In the latter, it was seen that exposure to ISO depressed force development by causing decreases in myofilament calcium ion ($Ca^{2+}$) responsiveness [9], possibly as a result of modification calmodulin affinity for the cation [10]. Further, in mice, ISO has been shown to cause decreased peripheral vascular resistance [11]. At least in rodent models where it has been evaluated, repeated administration of ISO results in more adverse effects than from just a single administration [12].

On the other hand, there are reports of beneficial 'side effects' from host exposures to ISO [13]. For example, it was reported that a volatile anesthetic like ISO could help protect the heart against ischemia-re-perfusion (I/R) injury [14, 15]; this is termed anesthetic-induced pre-conditioning (AIP). One study demonstrated that ISO pre-conditioning could reduce infarct size during I/R, in part, via some as of yet unknown regulation of cardiomyocyte death. Increasingly, these changes are thought to be due to alterations induced in the expression of a variety of micro-RNA (miRNA) in the tissues themselves [16–26]. Another study indicated that ISO exposure could reduce cell injury by imparting anti-inflammatory, anti-oxidative, and anti-apoptotic activities [27, 28].

Currently, it remains unknown if repeated acute exposure to ISO influences gene regulation in host heart tissues, especially those associated with heart failure and/or other cardiomyopathies. It is also not certain if such repeated exposures to ISO can lead to long-term changes that might be related to cellular damage and gene regulation. A recent study examined such changes in outbred Sprague-Dawley rats; however, those rats were anesthetized with a 2% ISO/98% air mixture for 90 min and then immediately euthanized to allow for their liver, kidney, and heart to be examined in rat oligonucleotide gene arrays [29]. That study found that liver showed the greatest number of differentially expressed genes (DEG) in response to ISO exposure, with 725/4900 (~ 15%) genes showing significant up-/down-regulation; the kidney had fewer anesthetic-regulated genes with 214 (~ 4%), compared to the heart that had the least alterations with 137 (~ 3%) immediately impacted by the ISO exposure.

To begin to address this uncertainty about risk to the heart from repeated acute exposures to ISO, groups of spontaneously hypertensive (SHR) rats were randomly selected and exposed on two consecutive days (2 hr/d, by intratracheal inhalation [ITIH]) to an air (97.5%)/ISO (2.5%) mixture. Control rats (naïve) received air only. Hearts were then harvested at multiple timepoints over a 1-yr period (Days 1, 30, 240, and 360 post-exposure) and total RNA extracted from the left ventricle (LV) for subsequent global gene expression analysis and compared with the randomized groups of unexposed naïve controls.

The data obtained herein showed that at all post-exposure timepoints, there were a multitude of DEG that appeared in the heart tissues in response to the ISO exposures. The data suggest to us that repeated acute ISO exposures may lead to initial significant impacts on genes associated with wound repair, inflammatory responses, and circadian rhythm regulation at early timepoints post-exposure, and that the magnitude of some of these changes diminish

over time. In contrast, in the case of long-term effects, the majority of DEG involved in extra-cellular matrix organization, cell signaling, cell differentiation, cell migration, cell-substrate adhesion, heart development, and regulation of blood pressure remained up-regulated by ISO on Day 360. In addition, analysis of DEG at Days 240 and 360 post-exposure indicated late onset DEG reflecting potentially long-lasting effects from repeated acute ISO exposures. Among these DEG were several associated with oxidative phosphorylation, ribosome, angio-genesis, mitochondrial translation elongation, focal adhesion among other ISO-impacted pathways.

## Materials and methods

### Animals

An SHR (spontaneously hypertensive rat) rat model was used for these studies. Selection of the SHR as the model was based on several factors: (1) it is a commonly-used model of chronic hypertension that progresses to heart failure during the last 6 mo of its lifespan of $\approx$ 2 years; (2) it can be used to study mechanisms of hypertension-induced hypertrophy as it progres-ses to heart failure; (3) it has been used successfully in studies pertaining to damage to arteries in general and athero-/arteriosclerosis in particular; and, (4) its well-documented use in studies to assess inducible changes in CV structure/function/gene expression [30, 31].

Male SHR (10-wk-old; Harlan Labs, Frederick, MD; n = 5-6/group/timepoint), were placed in polycarbonate cages with corncob bedding in a facility maintained at 23˚C with a 30–50% relative humidity and a 12-hr light/dark cycle. Food (Purina lab chow) and tap water were pro-vided *ad libitum*. All rats were acclimated 1 wk prior to use. All animal procedures were con-ducted under an animal protocol (Protocol Number: IA16-01467) approved by New York University Institutional Animal Care and Use Committee (IACUC).

### Experimental design

For the study, spontaneously hypertensive (SHR) rats were randomly selected and exposed for 2-hr periods on two consecutive days to isoflurane anesthesia (ISO; IsoFlo, Abbott Laborato-ries, North Chicago, IL) in $O_2$ carrier gas (2.5% final concentration after mixing with house air); the randomized naïve rats exposed to filtered air only were used as controls. All rats were exposed to the ISO in an intratracheal inhalation (ITIH) integrated system [32]. This particular method of exposure was required as ISO was needed to keep rats quiescent during experiments performed for exposures of parallel sets of rats to World Trade Center (WTC) dusts. No ani-mals in the study reported herein were exposed to the WTC-dust themselves.

Normal nose-only or whole-body exposures of the rats to the dusts were not plausible as the particles were predominantly ($>$ 95%) coarse-supercoarse in aerodynamic diameter and the rats would suffocate as their noses quickly plugged up with dust. The ITIH method allowed for a bypass of the nasal route and thereby direct introduction of the dust particles into the lungs. However, because the rats had to remain sedated during each exposure, it was key that any effects from the ISO itself on the heart tissues be identified so that distinct effects on the heart from the dusts themselves could subsequently be identified and quantified.

In brief, the rats were exposed to a 97.5% air/2.5% ISO mixture on two consecutive days (2 hr/d) by intratracheal inhalation via an instilled tube (a rounded-off tip of an 18-G Insyte). There was a total of two insertions of the Insyte, spaced 24 hr apart. The basic ITIH system used has been described in detail in a previous paper (Rivard et al. 2006) [33]; specific detailed protocols for all steps of the exposures used here were described in detail in previous papers [32, 34]. The naïve control SHR rats were collected in an age-matched fashion.

At various timepoints (Day1, 30, 240, and 360) post-exposures, rats from each treatment group were euthanized by injection with Sleepaway (500 mg/kg; Fort Dodge Animal Health, Fort Dodge, IA). At necropsy, heart tissues were removed and weighed; portions of each left ventricle (LV), left atrium (LA), right ventricle (RV), and right atrium (RA) were collected, flash frozen in liquid $N_2$, and placed at -80°C for later use. Only a portion of the left ventricle (LV) was used for the RNA-seq analyses performed in the current study.

## RNA-seq analysis

RNA-seq protocols routinely performed at our NIEHS center were used to analyze changes in the transcriptome (mainly including mRNA and long non-coding RNA) to identify key components targeted and the gene networks and pathways that may contribute to the adverse health effects/pulmonary pathologies associated with isoflurane induced pathways [35, 36]. In brief, total RNA from each rat LV was isolated using TRizol (Invitrogen, Waltham, MA). Any DNA contamination was removed using a DNA-free kit (Invitrogen) according to manufacturer protocols. From the final materials (purity and concentration confirmed using formaldehyde gels and a Nano UV-Vis Spectrophotometer), RNA-seq libraries were prepared using a TruSeq RNA Sample Prep kit v2 (Illumina). Briefly, polyA RNA were enriched using Oligo (dT) magnetic beads and chemically fragmented. First-strand cDNA synthesis was performed using random hexamer primer, and followed by the second-strand synthesis to create double-stranded cDNA. The blunt-end cDNA fragments were generated and an A-base was added into the blunt ends of each strand. The fragments were then ligated with the Multiplex adaptors (Illumina) and were ready for amplification and sequencing. Next, an Illumina HiSeq-4000 system (NYU Genome Technology Center) was used to perform sequencing of 1 x 50 single-end reads. Raw sequencing data (Fastq) were loaded into CLC Genomics Workbench (Version 20.0.4 Qiagen) for data analysis. The raw Fastq files were trimmed to remove any remaining adaptors and ambiguous nucleotides. The trimmed sequence files were aligned to human genome (Hg38) allowing two mismatches. Reads mapped to the exons of a gene were summed at the gene level. Gene expression levels were quantified as total counts per million (TPM).

## Quality assurance

Genes with low expression values (counts per million lower than 0.5 across all samples) were excluded from the subsequent analysis. The RNA-seq read counts data were normalized by the trimmed mean of M-values (TMM) method after filtering. Principal component analysis (PCA) was performed on normalized log-transformed expression values (after filtering) for ISO and Naïve samples to detect any possible outliers at each timepoint. Based on the PCA plots (S1 Fig) made from the first two principal components, both ISO and Naïve samples clustered together respectively in the PCA and there was no evidence of outliers among the samples. We have also tried to identify potential batch effects across different timepoints by plotting gene expressions measured at different timepoints from the same tissue (data not shown). No evidence of batch effects was found.

## Differential gene expression analysis and enrichment analysis

Differential gene expression analysis was conducted to compare the outcomes from the ISO vs naïve rat tissues at each post-exposure timepoint. Genes with very low expression levels were excluded from subsequent analyses in the quality assurance step. Differential gene expression analysis was performed using the R/Bioconductor software package edgeR in the R statistical programming environment [37, 38]. For each gene, the expression level was modeled by the

generalized linear model. Quasi-likelihood (QL) F-test was used to compare the gene expression value between the ISO group and naïve control group. The Benjamini-Hochberg method was used as an FDR adjustment for multiple testing correction. Heatmaps were generated using the R package pheatmap. Volcano plots were generated using the R package Enhanced-Volcano. To see if the top DEG were enriched in certain pathways, functional gene set enrichment analysis (GSEA) was conducted using metascape (http://www.metascape.org) [39].

## Statistical analysis

Differential gene expression analysis was performed at each timepoint with adjustment for multiple comparison using the Benjamini-Hochberg method to control false discovery rate (FDR). Based on the log fold-change (logFC) at each post-exposure timepoint, the longitudinal trajectory for each gene was created to examine short-term or long-term trends. Principal component analyses (PCA) were conducted on normalized log-transformed expression values from ISO and Naïve samples at each timepoint to examine clustering patterns and detect potential outliers. Distributions of logFC values on different days were tested using Mann-Whitney U test and Kruskal-Wallis test. The Jonckheere-Terpstra test was applied to test the trend of trajectories, i.e., to test whether the logFC values increase/decrease along time (against the null hypothesis of no trend), by comparing the median logFC values over time. PCA and all statistical tests were conducted in the R programming environment [37, 38].

## Results

### Differential gene expression

Rats (n = 5-6/group) were exposed on two consecutive days (2 hr/d, by intratracheal inhalation) to an air (97.5%)/ISO (2.5%) mixture, and their hearts were then harvested at post-exposure Days 1, 30, 240, and 360. Naïve rats exposed only to air were used as controls. The differential gene expression analysis results are summarized and illustrated in volcano plots (Figs 1 and 2). Fig 1 was generated by pooling results (ISO vs naïve) from all groups from each timepoint, while Fig 2 shows results from each group (ISO vs naïve) at single timepoints. Each volcano plot presents the nominal p-value against fold-change after log transformation for every gene represented by each dot. Red dots indicate differentially-expressed genes (DEG) with a p-value < 0.05; black dots indicate non-significant (NS) genes.

Fig 1 demonstrates there were a vast number of DEG and significant impact from the two acute ISO exposures. Furthermore, a significant impact of the ISO treatments on some genes persisted throughout the entire Day 1–360 post-exposure period (Fig 2). The large number of DEG between the ISO and naïve rats at Day 360 indicates potentially long-term effects on the heart from the anesthetic.

Interestingly, the number of ISO-induced DEG was less on Days 30 and 240 than on Days 1 or 360 (Fig 2). Presumably, while Day 1 data contains DEG that later dissipated, the Day 360 data contains a large number of DEG that seemed to be late onset at later timepoints post-exposure. In particular, among the Day 1 rats, there were multiple genes with strong and significant expression changes directly due to a stimulus from the acute repeated ISO exposures. A portion of the DEG observed on Day 1 appeared to be associated with the initial traumatic impact of the intratracheal inhalation. At Days 30 and 240, many wound-response genes and other Day 1 DEG associated with any initial "trauma" had dissipated or stabilized over time. Interestingly, longitudinal analyses indicated existence of a large number DEG that appeared to be late onset at the later timepoints (Day 240 and 360), reflecting potentially long-lasting effects from the repeated acute ISO exposures. Among Day 360 rats, most background noise genes and pathways that had appeared early on had dissipated, allowing for signals from genes

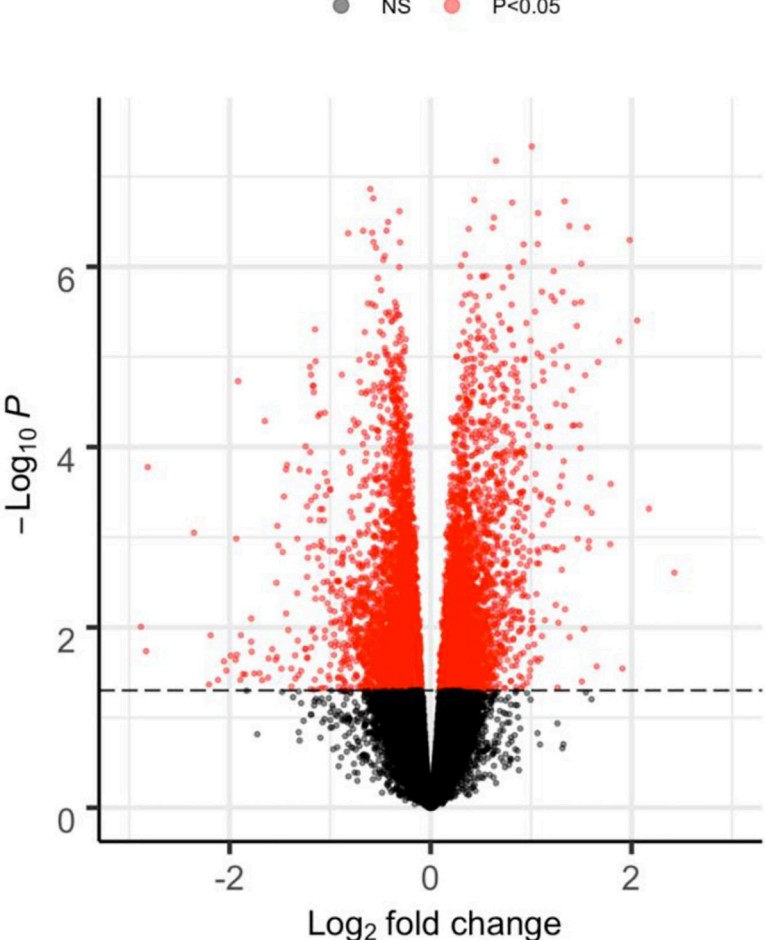

**Fig 1. Volcano plot pooling differential expression analysis results from all timepoints together (ISO vs. Naïve rats).** The volcano plot was generated by pooling differential gene expression analysis results (ISO vs naïve) from all post-exposure timepoints together. Groups of rats exposed to air (naïve, n = 6/each timepoint group) or ISO (n = 5/ Day 1 group, and n = 6/Day 30, 240, and 360 groups) were euthanized and their hearts removed for RNA harvest at Days 1, 30, 240, and 360 post-exposure. The volcano plot presents the nominal p-value against fold-change after log transformation for every gene represented by each dot. Red dots indicate differentially-expressed genes (DEG) with p-value < 0.05; black dots indicate non-significant (NS) genes.

associated with longer-term effects (including the late onset DEG) to become more evident/ detectable.

The heatmaps of the top 50 up-/down-regulated DEG at each timepoint are plotted in Fig 3 (Day 1 and 30) and Fig 4 (Day 240 and 360). These show that the variations of top DEG were primarily due to the group difference, outcomes consistent with the volcano plots.

## Short-term ISO effects on genes related to cardiac wound healing and circadian rhythm pathways

The early timepoint analyses revealed that ISO exposures impacted genes associated with wound healing and circadian rhythm pathways. What was not clear was if the magnitude of

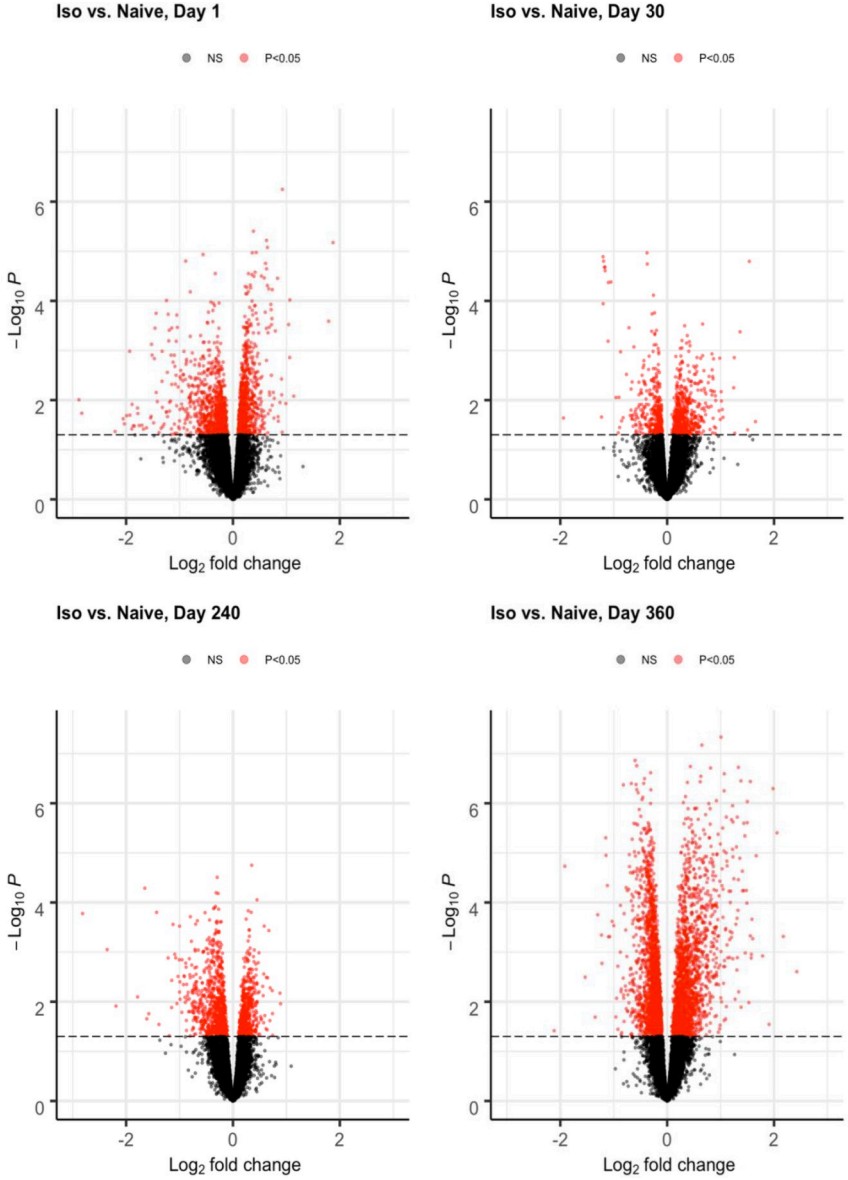

**Fig 2. Volcano plots of differential expression analysis (ISO vs. Naïve) at each timepoint.** The volcano plots were generated by pooling differential gene expression analysis results from each group (ISO vs naïve) at each post-exposure timepoint (Days 1, 30, 240, 360). Groups of rats exposed to air (naïve, n = 6/each timepoint group) or ISO (n = 5/Day 1 group, and n = 6/Day 30, 240, and 360 groups) were used. The volcano plot presents the nominal p-value against fold-change after log transformation for every gene represented by each dot. Red dots indicate differentially-expressed genes (DEG) with p-value < 0.05; black dots indicate non-significant (NS) genes.

these effects would persist or diminish over time. To obtain more clarity and an extensive view of potential effects from the ISO exposure, a list of 1846 genes from four Gene Ontology terms related to wound healing and circadian rhythm (i.e., response to wounding, detection of wounding, immune response, circadian rhythm regulation) were downloaded from QuickGo (https://www.ebi.ac.uk/QuickGO/). A total of 930 of the 1846 had measured gene expressions in the data. Of these, 122 showed significant differential expression due to ISO exposure on Day 1 post-exposure (i.e., 55 up-regulated, 67 down-regulated; significance was defined as p < 0.05). In addition to the genes that were significantly differentially-expressed on Day 1,

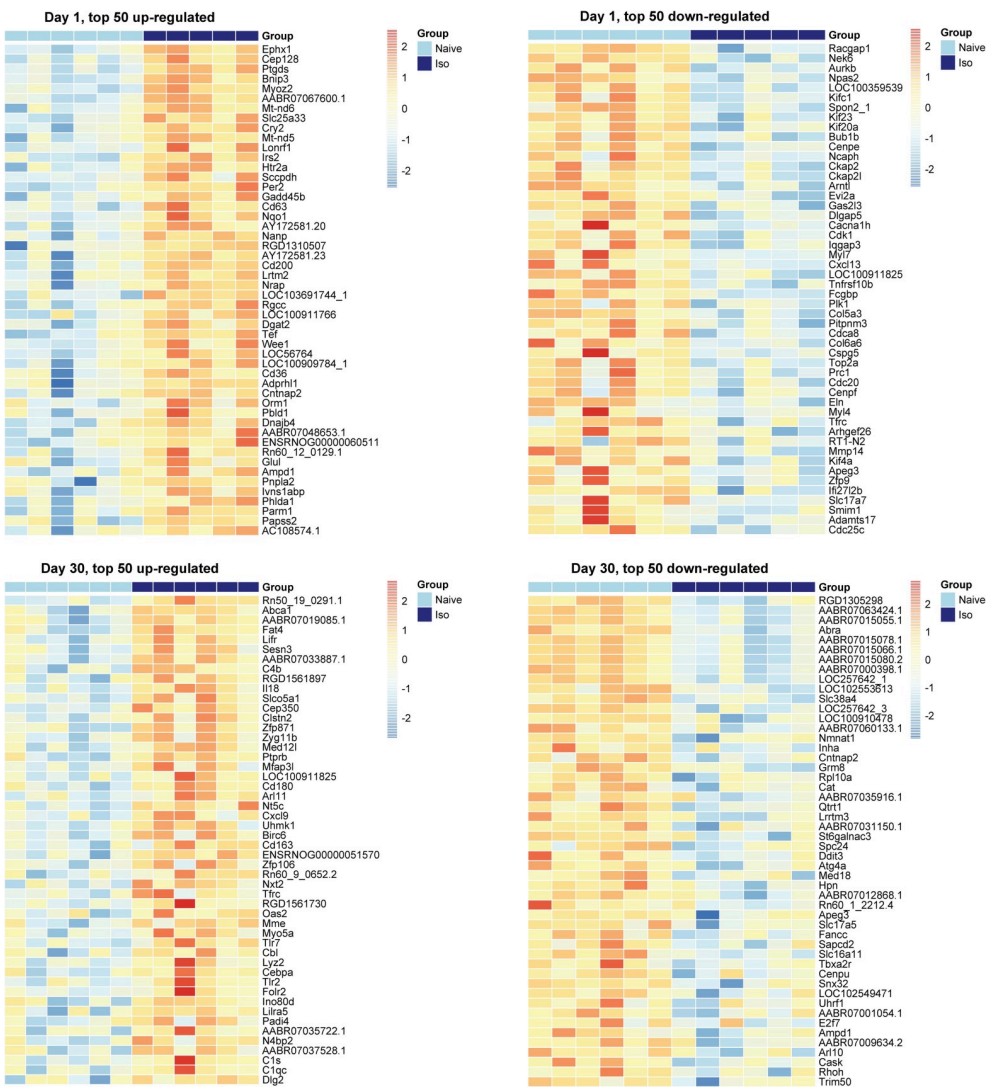

**Fig 3. Heatmaps of top 50 up-regulated and down-regulated DEG at Days 1, and 30.** In the group scale, the light blue stands for naïve samples and dark blue represents ISO samples. For each gene, expression values were standardized by the mean and standard deviation across samples. In the color scale, the red color means the expression value is above the average, while blue color means the expression value is below the average. Groups of rats exposed to air (naïve, n = 6 /Days 1 and 30 groups) or ISO (n = 5 /Day 1 group, and n = 6 /Day 30 group) were used.

the analyses then included genes that were changing in the same direction (between ISO treatment and naïve rats) with p-values < 0.1 on both Days 1 and 30 post-exposure. This process resulted in identification of 212 genes in total, with 87 being up-regulated and 125 down-regulated. The trajectories of the logFC among the 212 selected genes seen between the ISO and naïve rats were plotted and evaluated by mean/median on all days, and assessed using Mann-Whitney U test and Kruskal-Wallis test. The trajectory trends were also evaluated using Jonckheere-Terpstra test in up-/down-regulated genes, respectively. Lastly, functional enrichment analysis was performed to explain pathways that were being enriched by these 212 genes.

For each gene, at each post-exposure timepoint, the gene expression within each group can be averaged to derive the logFC between two groups of the rats, and then plot the logFC value vs. timepoint to form a trajectory for each gene. The trajectories of the logFC between the ISO-

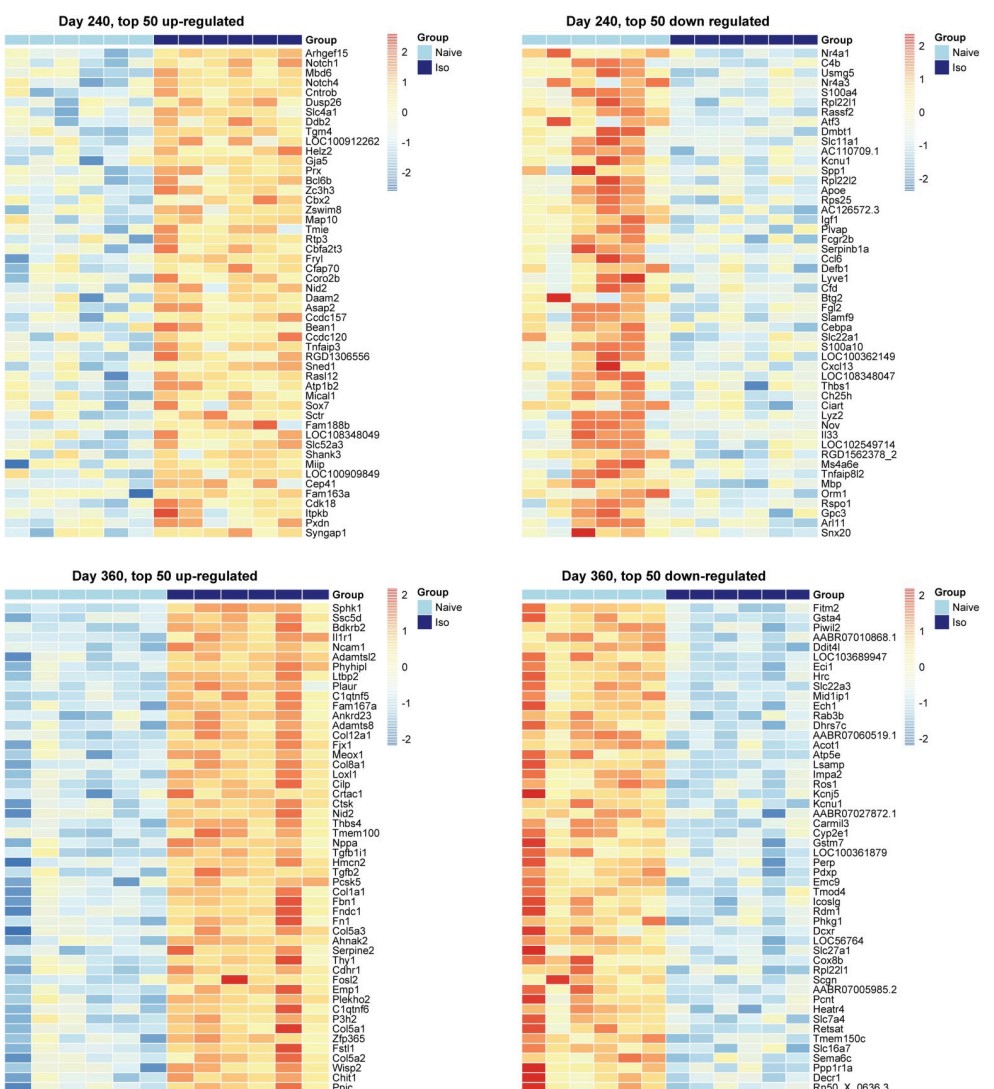

**Fig 4. Heatmaps of top 50 up-regulated and down-regulated DEG at Days 240, and 360.** In the group scale, the light blue stands for naïve samples and dark blue represents ISO samples. For each gene, expression values were standardized by the mean and standard deviation across samples. In the color scale, the red color means the expression value is above the average, while blue color means the expression value is below the average. Groups of rats exposed to air (naïve, n = 6/Day 240 and 360 groups) or ISO (n = 6/Day 240 and 360 groups) were used.

exposed and naïve rats for the 212 genes are plotted, respectively, in Figs 5 and 6 as gray lines; the blue lines connecting the blue triangles are the mean trajectory. Fig 5 displays the gene expression difference, i.e., logFC between ISO and naïve arm trajectories of the up-regulated genes defined on Day 1; Fig 6 displays trajectories of counterpart down-regulated genes. Based on the two figures, the average trajectory of these 212 DEG suggested to us that the systematic difference between ISO and naïve hearts was the largest on Day 1, decreasing slightly by Day 30, and approaching 0 by Day 240 and thereafter.

By comparing the median logFC values over time, the Jonckheere-Terpstra trend test showed there were significant decreasing trends [$p < 2 \times 10^{-16}$] for Day 1 up-regulated DEG on Day 1 (Fig 5) and increasing trends [$p < 2 \times 10^{-16}$] for the down-regulated DEG (Fig 6). Results of the Mann-Whitney U-test and the Kruskal-Wallis test suggested there was

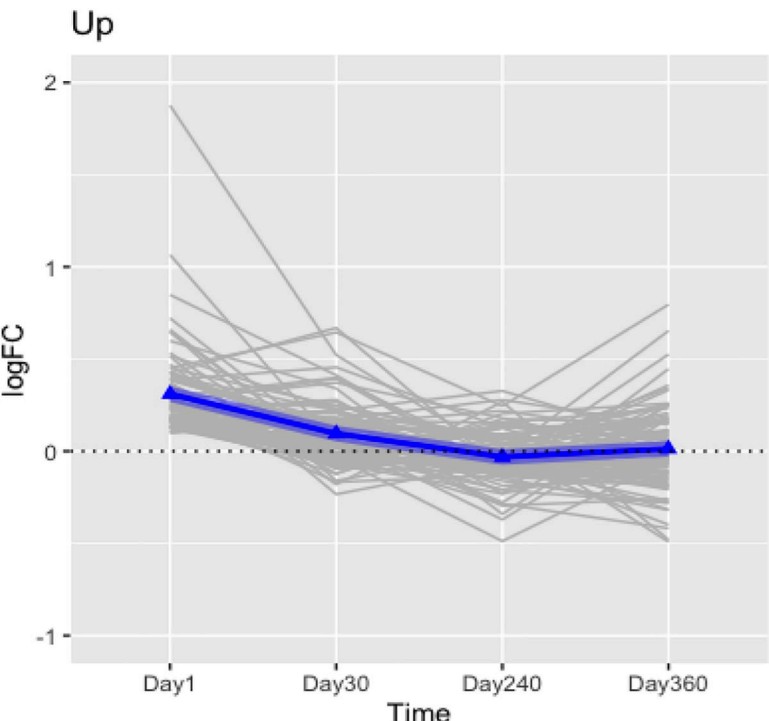

**Fig 5. Trajectories of logFC of the DEG up-regulated with short-term effects (ISO vs. naïve rats).** The trajectory plot was generated using the logFC value between the ISO and naïve rats vs each post-exposure timepoint for each up-regulated gene. The 87 DEG (either up-regulated with p-value < 0.05 on Day 1 or consistently up-regulated with p-value < 0.1 at Days 1 and 30) were associated with wound healing and circadian rhythm pathways. The trajectories of the logFC for each gene were plotted as gray lines. The blue lines connecting the blue triangles are the mean trajectory. Groups of rats exposed to air (naïve, n = 6/each timepoint group) or ISO (n = 5/Day 1 group, and n = 6/Day 30, 240, and 360 group) were used.

significant differences in logFC from Day 1 to Day 240 samples, but no statistical difference in logFC between Day 240 and Day 360 samples (Table 1). Functional enrichment analysis of these 212 genes confirmed wound healing was the most important pathway/process impacted in the left ventricle by ISO treatment early after conclusion of the exposures. This finding remained the same through Day 30; only by Day 240 did the impact fade away. Lastly, from the functional gene set enrichment analysis (GSEA), it can be seen the two acute exposures to ISO impacted on genes associated with regulation of circadian rhythm, (Fig 7). However, as with wound healing, the effects were most significant early-on post-exposure, and then diminished over time.

Table 2 summarized the top four most significant pathways enriched by the 212 DEG. However, the relatively small sample size of rats at each timepoint and the large number of genes tested lead to a typical multiple testing situation. A portion of genes with p-value < 0.05 are likely due to chance, even in specific pathways. Thus, false discovery rate (FDR) adjustment is reported in addition to unadjusted p-values. The large number of DEG with FDR < 0.05 suggests signal beyond noise and justifies a GSEA approach. The log-10 transformed nominal p-values and FDR for each pathway are reported in Table 2. A gene list generated from genes in the top four significant pathways (from GSEA) is reported in Table 3. For examples, Bnip3 (p = 8.40 x $10^{-6}$, FDR = 2.01 x $10^{-2}$), Adam17 (p = 2.88 x $10^{-5}$, FDR = 2.57 x $10^{-2}$), Cx3cl1 (p = 1.10 x $10^{-4}$, FDR = 4.33 x $10^{-2}$) are among the DEG in the response to wound pathways at Day 1. Ptgds (p = 6.86 x $10^{-6}$, FDR = 2.01 x $10^{-2}$), Cry2 (p = 3.16 x $10^{-5}$, FDR = 2.57 x $10^{-2}$),

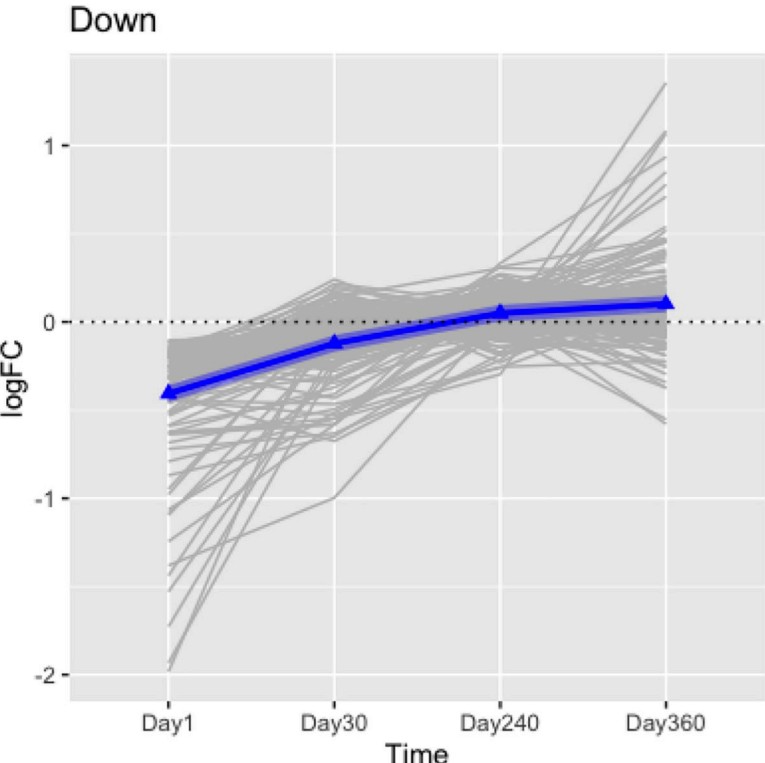

**Fig 6. Trajectories of logFC of the DEG down-regulated with short-term effects (ISO vs. Naïve).** The trajectory plot was generated using the logFC value between the ISO and naïve rats vs each post-exposure timepoint for each down-regulated gene. The 125 DEG (either down-regulated with p-value < 0.05 on Day 1 or consistently down-regulated with p-value < 0.1 at Days 1 and 30) were associated with wound healing and circadian rhythm pathways. The trajectories of the logFC for each gene were plotted as gray lines. The blue lines connecting the blue triangles are the mean trajectory. Groups of rats exposed to air (naïve, n = 6 /each timepoint group) or ISO (n = 5/Day 1 group, and n = 6/Day 30, 240, and 360 groups) were used.

Per2 (p = $9.57 \times 10^{-5}$, FDR = $4.28 \times 10^{-2}$) are among the DEG in the regulation of circadian rhythm pathway at Day 1. However, significance of these genes faded away at/after Day 30.

## Heatmaps for 212 DEG reflecting potential short-term effects over time

Heatmaps were generated for the 212 genes which were either in wound-related pathways with a p-value < 0.05 at Day 1, or having a p-value < 0.1 and changing in the same direction

**Table 1. Test results for means and trends of trajectories of the 212 genes associated with short-term effects from the repeated acute ISO exposures.**

| Null Hypothesis | Test | Alternative Hypothesis | p-value |
|---|---|---|---|
| Equal logFC distribution on Days 240 and 360 in the "Up" panel | Mann-Whitney U test | Two-sided | 0.233 |
| Equal logFC distribution on Days 240 and 360 in the "Down" panel | Mann-Whitney U test | Two-sided | 0.720 |
| Equal logFC distribution on Days 1, 30, and 240 in the "Up" panel | Kruskal-Wallis test | Two-sided | $< 2 \times 10^{-16}$ |
| Equal logFC distribution on Days 1, 30 and 240 in the "Down" panel | Kruskal-Wallis test | Two-sided | $< 2 \times 10^{-16}$ |
| No trend of logFC over time in the "Up" panel | Jonckheere-Terpstra trend test | Decreasing logFC trend over time | $< 2 \times 10^{-16}$ |
| No trend of logFC over time in the "Down" panel | Jonckheere-Terpstra trend test | Increasing logFC trend over time | $< 2 \times 10^{-16}$ |

The mean/median of trajectories on all days were assessed using Mann-Whitney U test and Kruskal-Wallis test. The trajectory trends were also evaluated using Jonckheere-Terpstra test.

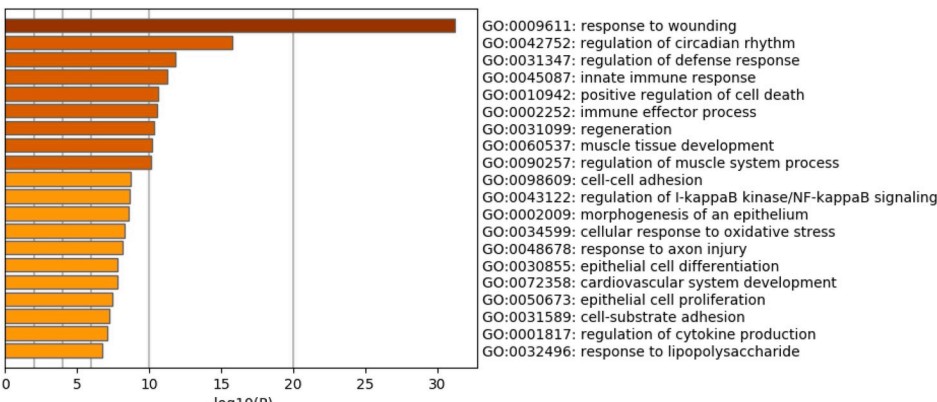

**Fig 7. Enrichment results of the 212 DEG reflecting short-term effects.** The figure presents the functional gene set enrichment analysis (GSEA) for the 212 selected DEG differentially-expressed between the ISO vs. naïve rats on Days 1 and 30. The X-axis is the negative log-transformed p-value of the enriched terms. The darker the color, the more statistically significant the enriched term (pathways, GO terms, etc.). Groups of rats exposed to air (naïve, n = 6/each timepoint group) or ISO (n = 5/Day 1 group, and n = 6/Day 30, 240, and 360 groups) were used.

at both Day 1 and Day 30 (logFC have same signs). Dividing these 212 genes into "up" and "down" according to their direction of regulation at Day 1 resulted in a total of 87 "up" genes and of 125 "down" genes. Among the 87 "up" genes, several were clearly up-regulated in ISO-treated group at Day 1, slightly up-regulated on Day 30, but the difference in expression continued to reduce over time until finally there was no significant difference in expression between ISO treated and naïve rats at Days 240 and 360 (Fig 8). Among the 125 "down" genes, several were clearly down-regulated on Day 1 and slightly down-regulated on Day 30. However, again, the difference in expression continued to reduce over time until finally there was no significant difference in expression between ISO exposed and naïve rats at Days 240 and 360 (Fig 9). These changing patterns of gene expression levels were consistent with the results shown in Figs 5 and 6.

## Differentially-expressed genes with potential long-lasting effects identified at late timepoints

Although the repeated acute ISO exposures affected genes significantly associated with wounding healing and circadian rhythm pathways at early post-exposure timepoints and this impact

**Table 2. Summary of top four significant pathways enriched by the 212 DEG reflecting short-term effects of repeated acute ISO exposures.**

| Term | Description | LogP | Log (FDR) | Gene symbols |
|------|-------------|------|-----------|--------------|
| GO:0009611 | response to wounding | -31.2333 | -26.967 | Adra2c, Tspo, Serpine1, Sod1, Dmd, Ccnb1, Tyro3, P2ry1, Cst3, Eno3, Msx2, F3, Smad3, Anxa5, Cd36, Smad4, Cspg5, Rab27a, Gata4, Adam17, Klf6, Ltbp1, Mertk, Anxa6, Txn2, Bnip3, Nol3, Col5a1, Cx3cl1, S100a9, Cdkn1a, Vwf, Rhoa, Kcnk2, Optn, Pdia6, Papss2, Il10rb, Uck2, Dsp, Ctnna1, Ap3b1, Itga5, Large1, Selenon, Scrib, Yap1, Trim72, Flrt3, Srf, Vash1, Akirin1, LOC100912195 |
| GO:0042752 | regulation of circadian rhythm | -15.7937 | -12.005 | Ppp1ca, Ptgds, Id2, Arntl, Clock, Per2, Per3, Cry2, Nr1d2, Rbm4, Spsb4, Prox1, Pspc1, Fbxl3, Thrap3, Npas2, Kdm2a, Btrc, Fbxw7 |
| GO:0031347 | regulation of defense response | -11.836 | -8.269 | Agt, C4bpb, Cd200, Abcc1, Serpine1, Sod1, RT1-A1, Tyro3, Smad3, Cd28, Cd36, Pla2g5, Hmgb2, Vamp3, Clock, Adar, Vamp7, Cx3cl1, S100a9, Optn, Nr1d2, Nlrc5, Foxp1, Pspc1, Alpk1, Npas2, Foxp3, Serpinb9, Ifnlr1, Traf3, Mef2c |
| GO:0045087 | innate immune response | -11.2816 | -7.970 | C4bpb, Gapdh, RT1-A1, Tyro3, Vnn1, Cd36, Hmgb2, Vamp3, Rab27a, Cited1, Adar, Prkd1, Vamp7, Cx3cl1, S100a9, Trim28, Prdx1, Ifi27, Nlrc5, Serinc3, Nrros, Gbp1, Pspc1, Ipo7, Alpk1, Serpinb9, Traf3, Trim13, LOC500956, Trim8, Rab20 |

**Table 3. The list of DEG in selected pathways reflecting short-term effects of repeated acute ISO exposures.**

| | | Day 1 | | | Day 30 |
| | Name | logFC | p-value | FDR | logFC |
|---|---|---|---|---|---|
| 1 | Ptgds | 1.88 | 6.86E-06 | 2.01E-02 | 0.52 |
| 2 | Bnip3 | 0.65 | 8.40E-06 | 2.01E-02 | -0.02 |
| 3 | Adam17 | -0.33 | 2.88E-05 | 2.57E-02 | 0.01 |
| 4 | Cry2 | 0.53 | 3.16E-05 | 2.57E-02 | -0.07 |
| 5 | Vnn1 | 0.33 | 4.72E-05 | 3.09E-02 | 0.04 |
| 6 | Per2 | 1.07 | 9.57E-05 | 4.28E-02 | 0.03 |
| 7 | Spsb4 | 0.36 | 9.84E-05 | 4.28E-02 | -0.01 |
| 8 | Cx3cl1 | -0.41 | 1.10E-04 | 4.33E-02 | 0.09 |
| 9 | Fbxw7 | -0.38 | 1.60E-04 | 5.05E-02 | 0.07 |
| 10 | Npas2 | -1.44 | 1.80E-04 | 5.23E-02 | -0.03 |
| 11 | Nol3 | 0.37 | 1.88E-04 | 5.34E-02 | -0.10 |
| 12 | Cd200 | 0.45 | 2.34E-04 | 6.00E-02 | 0.25 |
| 13 | Cd36 | 0.43 | 8.18E-04 | 1.01E-01 | 0.05 |
| 14 | Arntl | -1.93 | 1.03E-03 | 1.13E-01 | -0.25 |
| 15 | Smad3 | -0.30 | 1.65E-03 | 1.35E-01 | 0.01 |
| 16 | Kdm2a | -0.18 | 1.73E-03 | 1.35E-01 | 0.03 |
| 17 | Ifnlr1 | 0.37 | 1.74E-03 | 1.35E-01 | -0.12 |
| 18 | Id2 | -0.46 | 1.84E-03 | 1.37E-01 | 0.01 |
| 19 | Ccnb1 | -0.63 | 2.27E-03 | 1.46E-01 | -0.55 |
| 20 | Gapdh | -0.46 | 2.42E-03 | 1.49E-01 | -0.04 |
| 21 | Eno3 | -0.46 | 2.45E-03 | 1.49E-01 | -0.18 |
| 22 | Cspg5 | -0.98 | 2.66E-03 | 1.52E-01 | 0.08 |
| 23 | Tspo | 0.31 | 2.71E-03 | 1.53E-01 | 0.03 |
| 24 | Clock | -0.40 | 3.05E-03 | 1.60E-01 | 0.10 |
| 25 | Papss2 | 0.52 | 3.61E-03 | 1.72E-01 | -0.23 |
| 26 | Tyro3 | -0.26 | 4.21E-03 | 1.88E-01 | -0.07 |
| 27 | Alpk1 | 0.18 | 4.42E-03 | 1.89E-01 | -0.06 |
| 28 | Vamp3 | 0.16 | 4.81E-03 | 1.97E-01 | 0.04 |
| 29 | Btrc | -0.25 | 5.25E-03 | 1.99E-01 | 0.02 |

Top genes in selected pathways from Table 2. The unadjusted p-value and FDR at Day 1 and logFC at Days 1 and 30 are reported.

mostly dissipated over time, there were still a striking number of genes differentially expressed at Day 360 (see volcano plots in Figs 1 and 2). To explore the source of the differential expression at late timepoints, DEG that were consistently up-/down-regulated at both Day 360 with FDR < 0.05 and Day 240 with a p < 0.1 were selected. This process resulted in 345 genes, where 146 were consistently up-regulated and 199 consistently down-regulated at Days 240 and 360. The trajectories of the logFC among the 345 DEG between the ISO and naïve rats were plotted and the mean logFC on all days assessed by a Kruskal-Wallis test. The trajectory trends were also evaluated using a Jonckheere-Terpstra test in up-/down-regulated genes. Lastly, gene set enrichment analyses were performed to potentially define functional pathways enriched by the selected DEG.

Fig 10 presents the trajectory of logFC between ISO and naïve arm of the 146 DEG consistently up-regulated on Days 240 and 360. Fig 11 displays trajectories of the 199 DEG consistently down-regulated at these later timepoints. In general, the average trajectory of the 345 DEG suggested to us that the systematic difference between ISO and naïve rats was ≈ 0 at

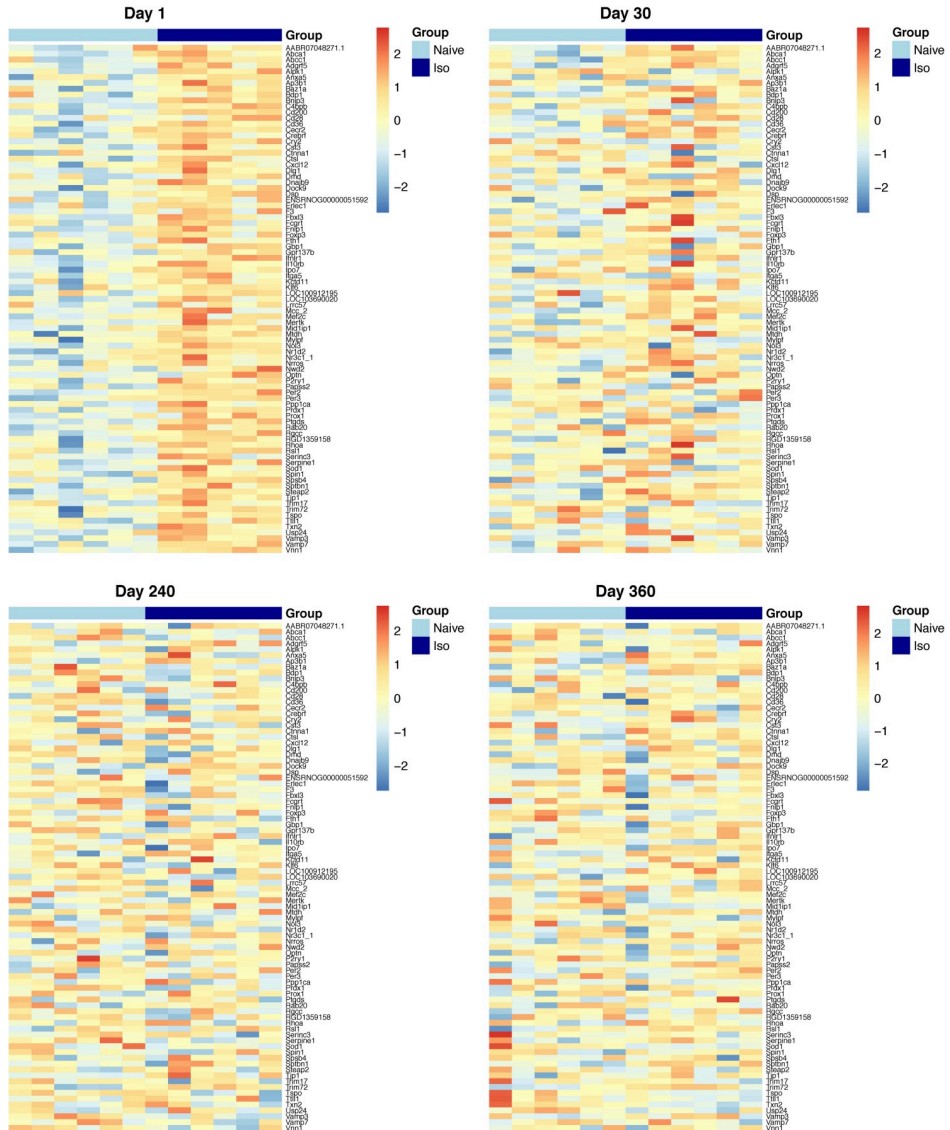

**Fig 8. Heatmaps of the 87 "up" genes reflecting short-term effects of repeated acute ISO exposures.** Heatmaps generated for the 87 up-regulated genes which were either in wound-related pathways with a p < 0.05 at Day 1, or having a p < 0.1 and changing in the same direction at both Day 1 and Day 30. Top left shows Day 1, top right shows Day 30, bottom left shows Day 240, and bottom right shows Day 360. The light blue represents samples from the naïve group; dark blue represents samples from the ISO group. Groups of rats exposed to air (naïve, n = 6/each timepoint group) or ISO (n = 5/Day 1 group, and n = 6/Day 30, 240, and 360 groups) were used.

Days 1 and 30, increasing after Day 30, and reaching a peak at Day 360. Thus, these DEG appear to represent late onset genes at these later timepoints, and so reflect potential long-lasting effects arising from repeated acute ISO exposures.

By comparing the median logFC values over time, the Jonckheere-Terpstra trend test showed there were significant increasing trends [p < 2 x 10$^{-16}$] for 146 consistently up-regulated DEG (Fig 10) and decreasing trends [p < 2 x 10$^{-16}$] for the 199 consistently down-regulated DEG (Fig 11) defined from Days 240 and 360. Kruskal-Wallis test suggested there was a significant difference between logFC values between the Day 30 to Day 360 samples (Table 4).

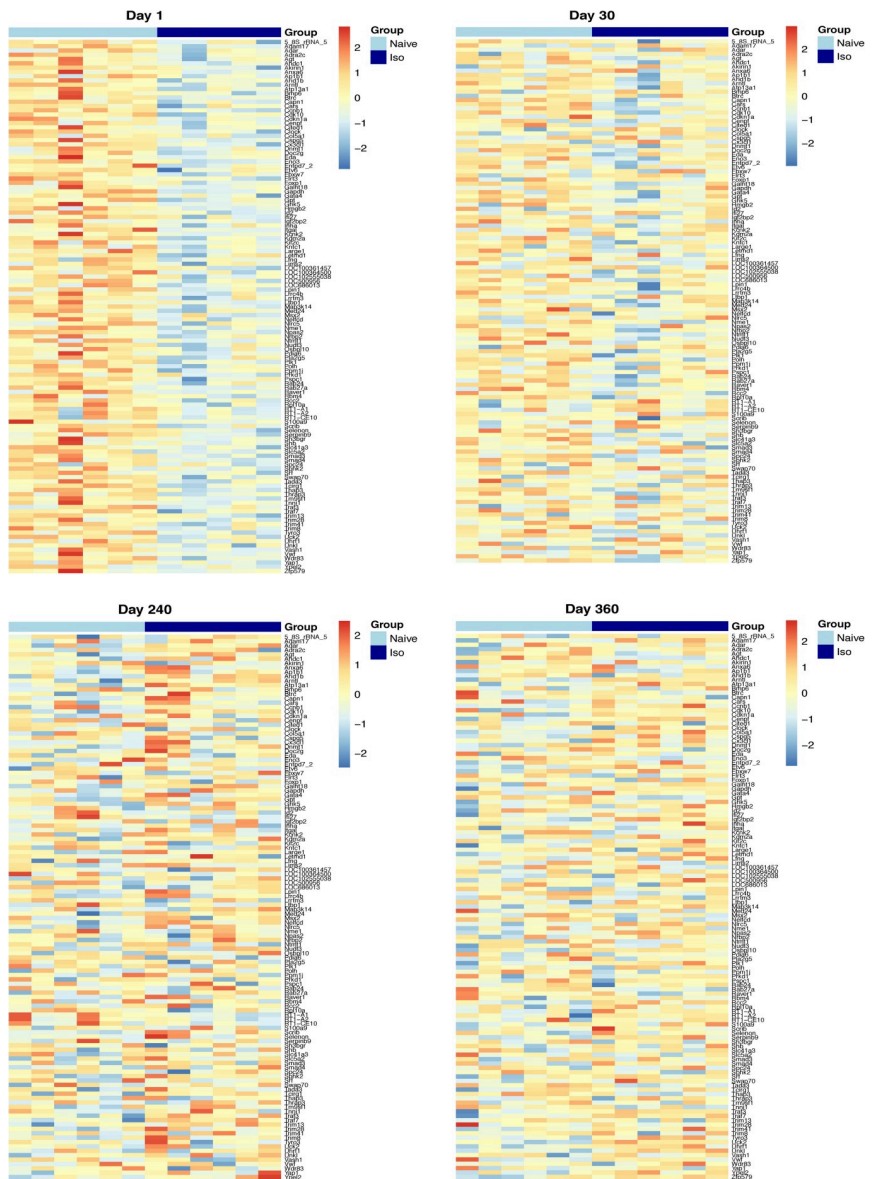

**Fig 9. Heatmaps of the 125 "down" genes reflecting short-term effects of repeated acute ISO exposures.** Heatmaps generated for the 125 down-regulated genes which were either in wound-related pathways with a p < 0.05 at Day 1, or a p < 0.1 and changing in same direction at both Day 1 and Day 30. Top left shows Day 1, top right shows Day 30, bottom left shows Day 240, and bottom right shows Day 360. The light blue represents samples from the naïve group; dark blue represents samples from the ISO group. Groups of rats exposed to air (naïve, n = 6/each timepoint group) or ISO (n = 5/Day 1 group, and n = 6/Day 30, 240, and 360 groups) were used.

Functional gene set enrichment analysis (GSEA) performed on the 345 long-term ISO induced genes shows these DEG were associated with several pathways, including oxidative phosphorylation, ribosome, angiogenesis, and mitochondrial translation elongation (Fig 12). These effects did not appear (significantly) until Day 240 and lasted through Day 360. The significant trend from the longitudinal analysis indicates potentially long-term effects arising from repeated acute ISO exposures.

Table 5 summarizes the top five significant pathways enriched by the 345 DEG affected by the ISO exposures at the later timepoints. Using the FDR adjustment for the multiple testing, a

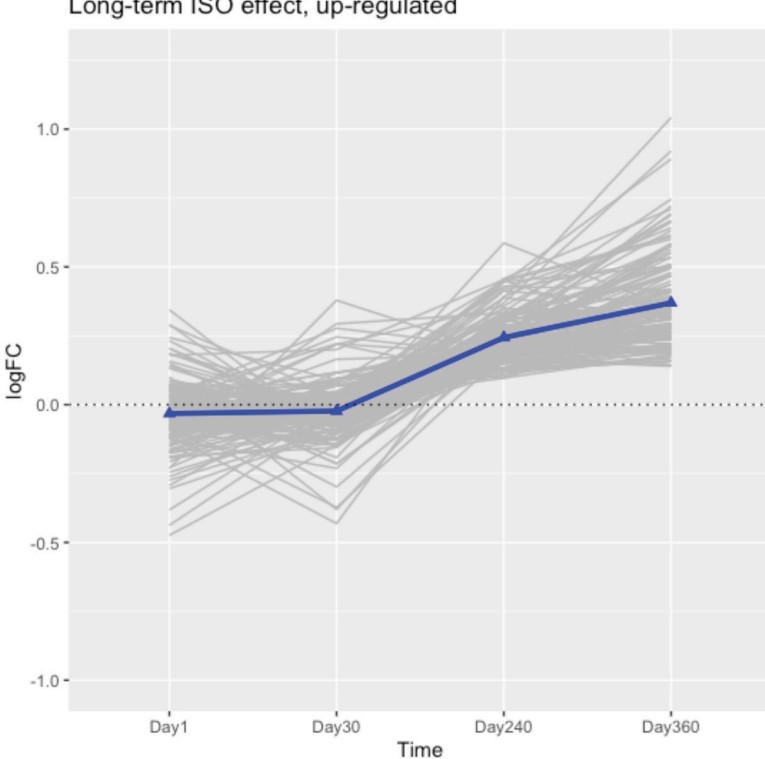

**Fig 10. Trajectories of logFC of significantly up-regulated genes that manifest at the late timepoints.** The 146 consistently up-regulated genes have FDR adjusted p-values < 0.05 at Day 360 and p < 0.1 at Day 240. Trajectories of individual genes plotted as grey lines. Blue lines connecting blue triangles are the mean trajectory. Groups of rats exposed to air (naïve, n = 6/each timepoint group) or ISO (n = 5/Day 1 group, and n = 6/Day 30, 240, and 360 groups) were used.

gene list generated from genes associated with the top five significant pathways (with FDR < 0.01) at Day 360 could possibly suggest the existence of signal beyond chance (Table 6). As examples, Ndufs5 (p = 1.07 x 10$^{-5}$, FDR = 1.06 x 10$^{-3}$), Atp5fl (p = 1.12 x 10$^{-5}$, FDR = 1.06 x 10$^{-3}$), Cox7b (p = 1.25 x 10$^{-5}$, FDR = 1.12 x 10$^{-3}$) were among the DEG in the oxidative phosphorylation pathway at Day 360; Col4a2 (p = 2.34 x 10$^{-7}$, FDR = 2.78 x 10$^{-4}$), Prkd2 (p = 3.43 x 10$^{-4}$, FDR = 2.78 x 10$^{-4}$), and Col4a1 (p = 1.40 x 10$^{-6}$, FDR = 4.54 x 10$^{-4}$) were among the DEG in the angiogenesis pathway at Day 360. There was no significance regarding these genes' regulations at Days 1 and 30.

## Heatmaps over time for the top DEG that appeared to be late onset

Heatmaps were generated for the top 50 late onset DEG, which were consistently up-/down-regulated DEG on Days 240 and 360 (Figs 13 and 14, respectively), at each timepoint. Fig 13 shows that the top 50 up-regulated DEG were not up-regulated in ISO until Days 240 and 360. Fig 14 shows that the top 50 down-regulated genes were not down-regulated in ISO group until Days 240 and 360. These patterns were supported by the longitudinal trajectory plots (see Figs 10 and 11).

## Functional gene set enrichment analysis (GSEA) at each timepoint

The top genes increased and decreased in the exposed hearts compared to controls at each of the different timepoints examined are listed in S1–S8 Tables. A GSEA review of these top

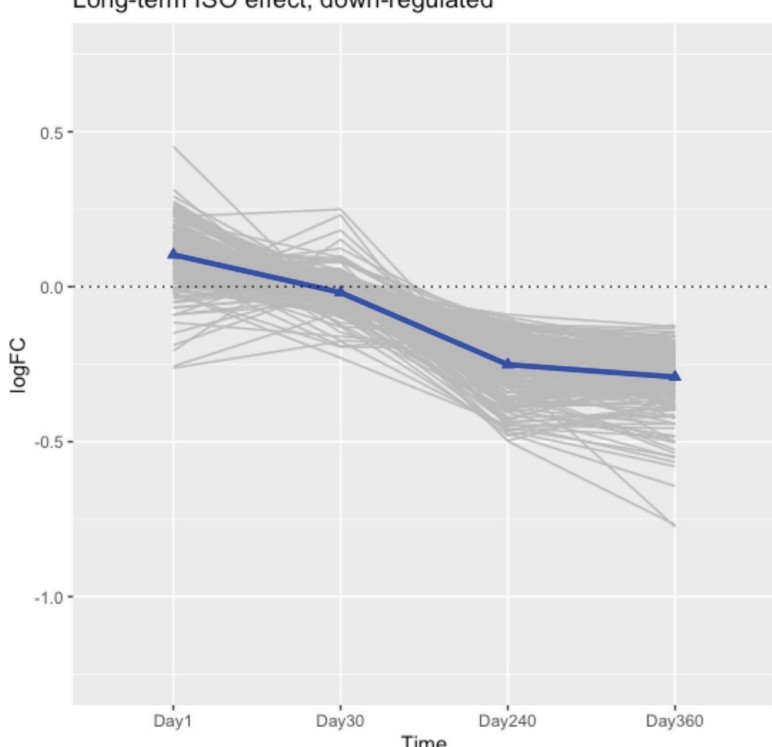

**Fig 11. Trajectories of logFC of significantly down-regulated genes that manifest at the late timepoints.** The 199 consistently down-regulated genes have FDR adjusted p-values < 0.05 at Day 360 and p < 0.1 at Day 240. Trajectories of individual genes plotted as grey lines. Blue lines connecting blue triangles are the mean trajectory. Groups of rats exposed to air (naïve, n = 6/each timepoint group) or ISO (n = 5/Day 1 group, and n = 6/Day 30, 240, and 360 groups) were used.

genes indicated that on Day 1, the repeated acute ISO exposures had imparted a significant up-regulating influence on genes related to lipid metabolism, including those associated with lipid storage, regulation of lipid localization, fat digestion/absorption, regulation/generation of precursor metabolites and regulation of energy and lipid biosynthetic processes (Fig 15). In addition, genes associated with small molecule biosynthesis, reactive oxygen species metabolic processes, and responses to mechanical stimulus were also up-regulated by the ISO treatments on Day 1. By Day 30, several genes involved in the immune response, including innate immunity, immune effector processes, myeloid leukocyte activation, inflammatory responses, macrophage differentiation, leukocyte proliferation, defense responses to exogenous organisms, and cytokine-mediated signaling pathways were all up-regulated as a result of the repeated acute ISO exposures. By Day 240, multiple genes involved in regulation of systemic arterial

**Table 4. Test results for means and trends of trajectories of the 345 late onset genes.**

| Null Hypothesis | Test | Alternative Hypothesis | p-value |
|---|---|---|---|
| Equal logFC distribution on Days 30, 240, and 360 in the "Up" panel | Kruskal-Wallis test | Two-sided | $< 2 \times 10^{-16}$ |
| Equal logFC distribution on Days 30, 240 and 360 in the "Down" panel | Kruskal-Wallis test | Two-sided | $< 2 \times 10^{-16}$ |
| No trend of logFC over time in the "Up" panel | Jonckheere-Terpstra trend test | Increasing logFC trend over time | $< 2 \times 10^{-16}$ |
| No trend of logFC over time in the "Down" panel | Jonckheere-Terpstra trend test | Decreasing logFC trend over time | $< 2 \times 10^{-16}$ |

The mean/median of trajectories were assessed using a Kruskal-Wallis test. The trajectory trends were evaluated using a Jonckheere-Terpstra test.

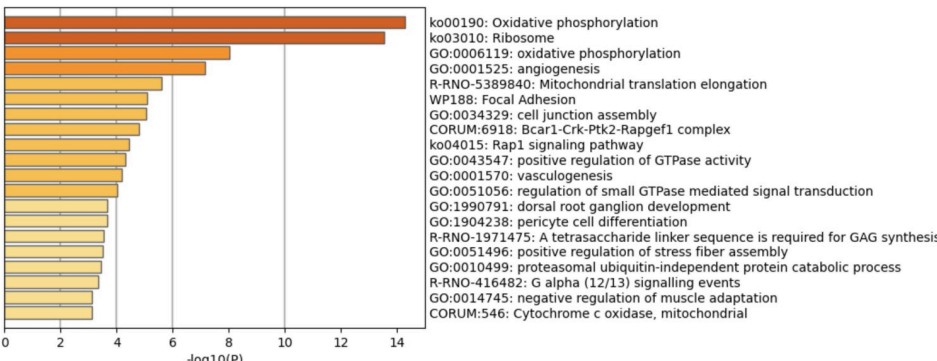

**Fig 12. Enrichment result of the 345 DEG that manifest at the late timepoints.** This figure presents the GSEA results for the 345 consistently up-/down-regulated DEG defined between the ISO vs. naïve rats at Days 240 and 360. The X-axis is the negative log-transformed p-value of the enriched terms. The darker the color, the more statistically significant the enriched term (pathways, GO terms, etc.). Groups of rats exposed to air (naïve, n = 6/each timepoint group) or ISO (n = 5/Day 1 group, and n = 6/Day 30, 240, and 360 groups) were used.

blood pressure, artery morphogenesis, and cell signaling pathways were now up-regulated due to the ISO exposures. By Day 360, genes involved in extracellular matrix organization, cell signaling, cell differentiation, cell migration, cell-substrate adhesion, heart development, and regulation of blood pressure were now seen to have been up-regulated as a result of the ISO exposures.

Short-term ISO treatment also led to significant down-regulation of genes. At Day 1 post-exposures, there was significant ISO-induced down-regulation of genes related to cell cycle pathways including those associated with sister chromatid segregation, cell cycle checkpoints, meiotic cell cycle processes, meiotic cell cycle processes, SUMOylation of DNA replication proteins, and microtubule bundle formation (Fig 16). By Day 30, the majority of down-regulated genes were still found to be involved in cell cycle pathways, including some related to amplification of signals from kinetochores, establishment of organelle localization, cell division, and cellular component disassembly. By Day 240, the majority of down-regulated genes were involved in the immune response (e.g., inflammation, cytokine production, tumor necrosis factor responsivity, myeloid leukocyte activation, responses against exogenous organisms,

**Table 5. Summary of top five significant pathways enriched by the 345 late onset DEG.**

| Term | Description | LogP | Log (FDR) | Gene symbols |
|---|---|---|---|---|
| ko00190 | Oxidative phosphorylation | -14.3014 | -10.323 | Ndufa5, Ndufs6, Cox7a2, Cox6c, Ndufv2, Atp5j, Atp5f1, Atp5e, Ndufa2, Ndufc2, Ndufab1, Atp5l, Cox7b, Ndufa6, Ndufs5, Ndufa1, Uqcr10, Ndufc1, Atp5j2, Cox7c, LOC100363268 |
| ko03010 | Ribosome | -13.5623 | -9.885 | Rps29, Rps14, Rps3a, Rpl35al1, Rpl30, Rpl36al, Rps27, Rps16, Rpl26, Mrps17, Mrps14, Mrps18c, mrpl11, Rpl7, Mrpl15, Rpl34, Rpl11, Rpl38, LOC100360449, Rpl37a, LOC100365810, LOC108352650 |
| GO:0006119 | oxidative phosphorylation | -8.01961 | -5.102 | Ndufs6, Cox7a2, Cox6c, Ndufv2, Atp5j, Atp5f1, Atp5e, Dnajc15, Ndufc2, Atp5l, Cox7b, Uqcr10, Atp5j2, Cox7c |
| GO:0001525 | angiogenesis | -7.15053 | -4.318 | Igf2, Nos3, Pdgfrb, Tbxa2r, Notch1, Ptk2, Smad1, Atp2b4, Sh2b3, Rapgef3, Nrp2, Cspg4, Ctnnb1, Tie1, Tgfbi, Cdh13, Ccl24, Col4a1, Prkd2, Rasip1, Rnf213, Col4a2, Cdh5, Epha2, Notch4, Vash2, Nrarp |
| R-RNO-5389840 | Mitochondrial translation elongation | -5.59928 | -2.953 | Mrps17, Mrps14, Mrps18c, mrpl11, Mrpl15, Mrpl54, Chchd1, Mrpl50, LOC103691922 |

The log-10 transformed nominal p-value and FDR-adjusted p-value for each pathway are reported.

**Table 6. Late onset DEG (identified at later timepoints) from significant pathways.**

|  | Name | Day 240 | | Day 360 | | |
|---|---|---|---|---|---|---|
|  |  | logFC | P value | logFC | P value | FDR |
| 1 | Col4a2 | 0.30 | 2.16E-02 | 0.63 | 2.34E-07 | 2.78E-04 |
| 2 | Prkd2 | 0.30 | 1.18E-02 | 0.38 | 3.43E-07 | 2.78E-04 |
| 3 | Col4a1 | 0.23 | 4.05E-02 | 0.72 | 1.40E-06 | 4.54E-04 |
| 4 | Tie1 | 0.27 | 4.24E-02 | 0.40 | 1.67E-06 | 5.20E-04 |
| 5 | Atp5e | -0.39 | 1.70E-03 | -0.50 | 5.50E-06 | 7.98E-04 |
| 6 | mrpl11 | -0.11 | 8.29E-02 | -0.26 | 1.01E-05 | 1.06E-03 |
| 7 | Mrps18c | -0.38 | 9.46E-04 | -0.34 | 1.06E-05 | 1.06E-03 |
| 8 | Ndufs5 | -0.20 | 9.14E-03 | -0.29 | 1.07E-05 | 1.06E-03 |
| 9 | Atp5f1 | -0.16 | 2.11E-02 | -0.28 | 1.12E-05 | 1.06E-03 |
| 10 | Cox7b | -0.46 | 9.33E-04 | -0.40 | 1.25E-05 | 1.12E-03 |
| 11 | Chchd1 | -0.39 | 1.58E-03 | -0.37 | 1.90E-05 | 1.40E-03 |
| 12 | LOC103691922 | -0.18 | 1.09E-02 | -0.36 | 2.50E-05 | 1.57E-03 |
| 13 | Cox7a2 | -0.27 | 4.96E-03 | -0.38 | 3.08E-05 | 1.74E-03 |
| 14 | Mrpl15 | -0.15 | 3.24E-02 | -0.25 | 3.10E-05 | 1.74E-03 |
| 15 | Ndufs6 | -0.13 | 6.30E-02 | -0.30 | 3.12E-05 | 1.74E-03 |
| 16 | Atp5l | -0.32 | 5.62E-04 | -0.35 | 3.25E-05 | 1.77E-03 |
| 17 | Atp5j2 | -0.27 | 1.95E-04 | -0.27 | 3.87E-05 | 1.99E-03 |
| 18 | Cdh13 | 0.24 | 4.64E-02 | 0.41 | 4.21E-05 | 2.09E-03 |
| 19 | Ndufc2 | -0.17 | 3.70E-03 | -0.20 | 5.49E-05 | 2.37E-03 |
| 20 | Smad1 | 0.13 | 4.67E-02 | 0.33 | 6.75E-05 | 2.66E-03 |
| 21 | Nos3 | 0.28 | 1.46E-02 | 0.33 | 7.17E-05 | 2.73E-03 |
| 22 | Atp5j | -0.11 | 7.55E-02 | -0.23 | 7.44E-05 | 2.78E-03 |
| 23 | Epha2 | 0.24 | 9.05E-02 | 0.50 | 7.54E-05 | 2.78E-03 |
| 24 | Mrpl50 | -0.23 | 1.89E-03 | -0.25 | 1.06E-04 | 3.48E-03 |
| 25 | Ndufa2 | -0.27 | 2.75E-03 | -0.29 | 1.11E-04 | 3.60E-03 |
| 26 | Ndufv2 | -0.14 | 7.19E-02 | -0.25 | 1.15E-04 | 3.64E-03 |
| 27 | Nrarp | 0.25 | 9.08E-02 | 0.69 | 1.23E-04 | 3.80E-03 |
| 28 | Vash2 | 0.16 | 5.81E-02 | 0.47 | 1.45E-04 | 4.26E-03 |
| 29 | Atp2b4 | 0.18 | 7.19E-02 | 0.58 | 1.67E-04 | 4.69E-03 |
| 30 | Mrpl54 | -0.19 | 8.66E-02 | -0.31 | 1.83E-04 | 4.95E-03 |
| 31 | Cox7c | -0.33 | 1.26E-02 | -0.34 | 2.18E-04 | 5.49E-03 |
| 32 | Dnajc15 | -0.30 | 9.62E-03 | -0.37 | 3.09E-04 | 6.81E-03 |
| 33 | Rasip1 | 0.22 | 1.48E-02 | 0.31 | 3.38E-04 | 7.16E-03 |
| 34 | LOC100360449 | -0.50 | 1.57E-03 | -0.36 | 3.61E-04 | 7.39E-03 |
| 35 | Rps3a | -0.31 | 2.35E-04 | -0.24 | 3.87E-04 | 7.73E-03 |
| 36 | Ndufa5 | -0.23 | 3.44E-03 | -0.23 | 4.55E-04 | 8.61E-03 |
| 37 | Rpl30 | -0.37 | 6.50E-04 | -0.26 | 4.96E-04 | 9.10E-03 |
| 38 | Ndufa1 | -0.20 | 2.00E-02 | -0.25 | 5.14E-04 | 9.29E-03 |

Genes were selected from Table 5 with FDR < 0.01 at Day 360. LogFC, as well as unadjusted p-values at Days 240 and 360 and FDR at Day 360 are shown.

innate immunity, and acute inflammatory responses). By Day 360, the majority of down-regulated genes were seen to be involved in fatty acid metabolism, including fatty acid metabolic processes, long-chain fatty acid metabolic processes, drug metabolism-cytochrome $P_{450}$ pathways, cofactor metabolic process pathways, and regulation of cellular component size pathways.

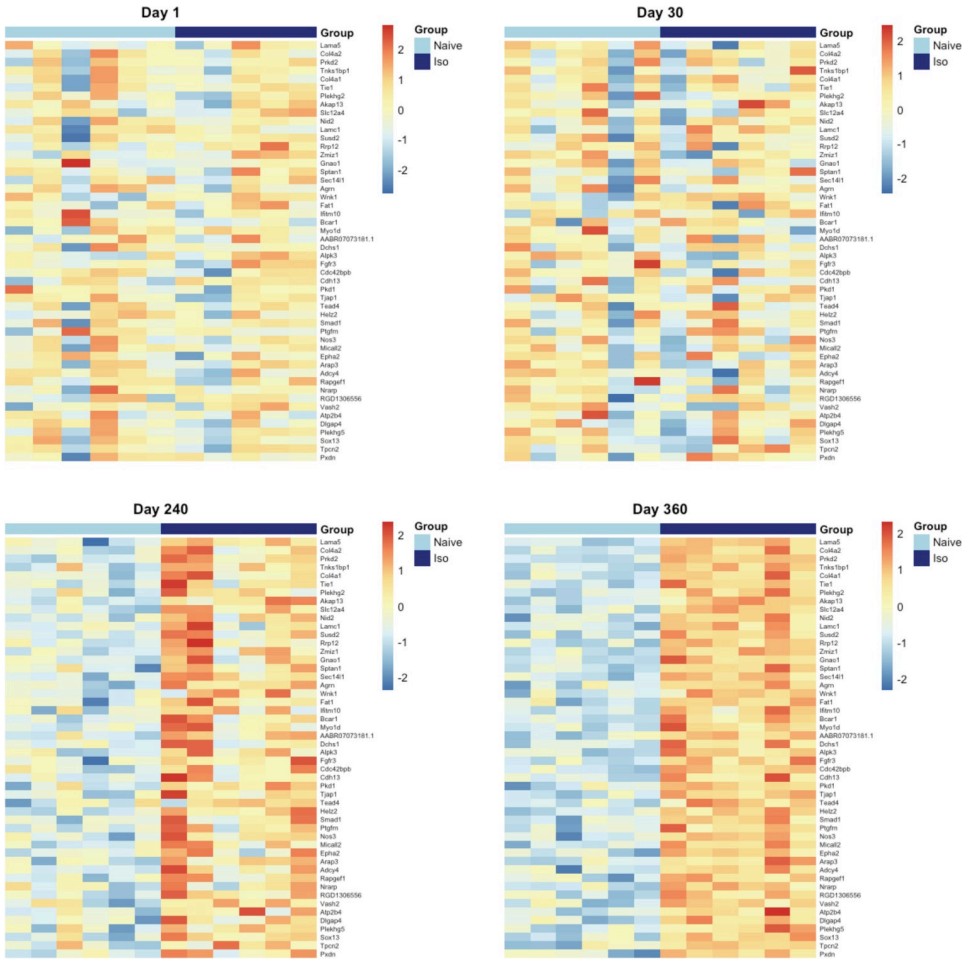

**Fig 13. Heatmaps of the top 50 consistently up-regulated DEG that were identified at the later timepoints.**
Heatmaps were generated for top 50 consistently up-regulated DEG according to FDR at Day 360. Normalized
expression levels were used and standardized within each gene. For the group scale, light blue represents naïve group
samples and dark blue represents ISO group samples. Groups of rats exposed to air (naïve, n = 6/each timepoint
group) or ISO (n = 5/Day 1 group, and n = 6/Day 30, 240, and 360 groups) were used.

## Discussion

In the present study, male spontaneously hypertensive rats (SHR) were exposed on two consec-
utive days (2 hr/d) to isoflurane (ISO) anesthetic and then their hearts were assessed for
changes in gene expression at multiple timepoints over a 1-yr period post exposures. From the
global gene expression analyses performed, at Days 1 and 30 post-exposure, genes associated
with processes pertaining to wounding, local immune function, inflammation, and circadian
rhythm regulation in the heart were significantly impacted by the repeated acute ISO expo-
sures. However, these specific gene effects disappeared by Days 240 and 360, indicating they
were not persistent. On the other hand, at Day 360 there were significantly increased numbers
of other DEG induced in the heart by the two repeated acute ISO exposures. Among these
were several associated with cell signaling, differentiation, and migration, extracellular matrix
organization, cell-substrate adhesion, heart development, and regulation of blood pressure. By
the final timepoint, a majority of the significantly down-regulated genes were involved in fatty
acid metabolism (including fatty acid metabolic processes, and both long-chain fatty acid

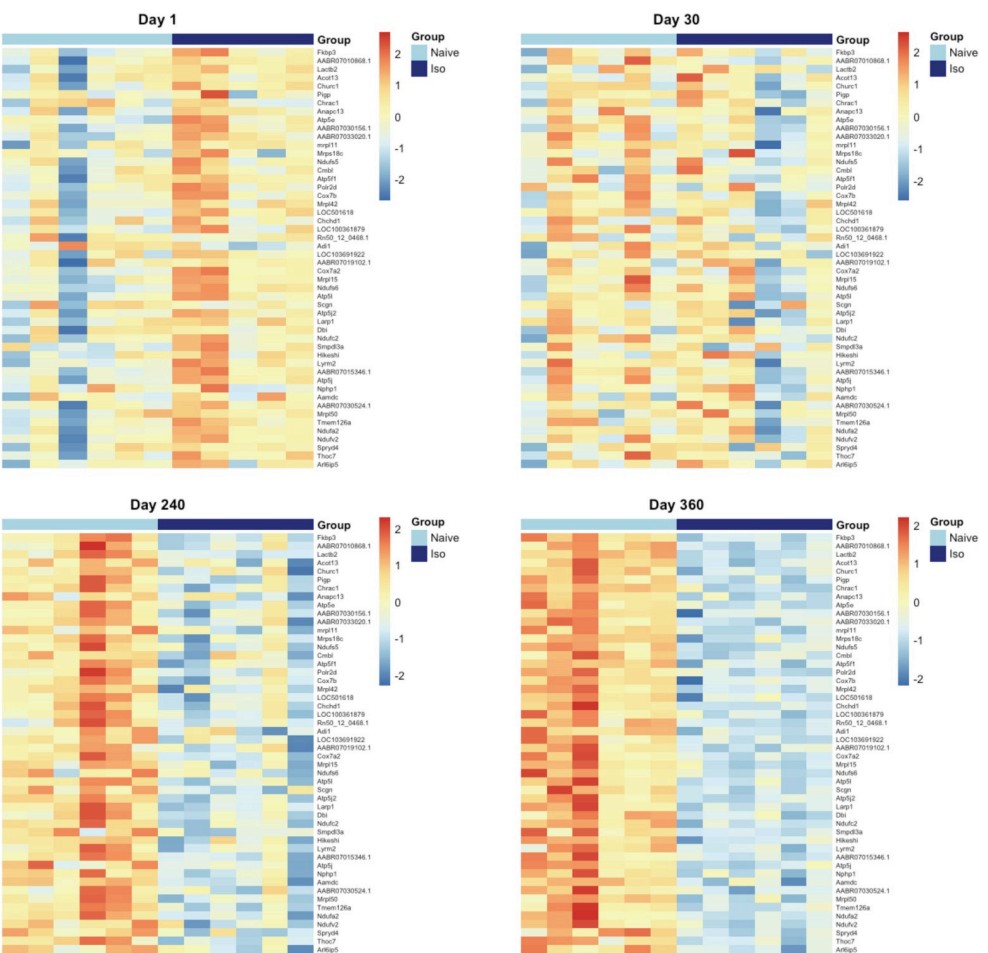

**Fig 14. Heatmaps of the top 50 consistently down-regulated DEG that were identified at the later timepoints.**
Heatmaps were generated for top 50 consistently down-regulated DEG according to FDR at Day 360. Normalized
expression levels were used and standardized within each gene. For the group scale, light blue represents naïve group
samples and dark blue represents ISO group samples. Groups of rats exposed to air (naïve, n = 6/each timepoint
group) or ISO (n = 5/Day 1 group, and n = 6/Day 30, 240, and 360 groups) were used.

metabolic processes), as well as related to component cytochrome P$_{450}$ pathways, cofactor metabolic process pathways, and regulation of cellular component size pathways. In addition, the longitudinal analysis of consistent DEG at Days 240 and 360 post-exposure indicated the presence of late onset DEG that potentially could reflect long-term effects from the repeated acute ISO exposures. These included DEG associated with a variety of important pathways, including oxidative phosphorylation, ribosome, angiogenesis, mitochondrial translation elongation, and focal adhesion.

ISO is a commonly-used inhalational anesthetic in clinical/surgical settings and in animal experimental studies. Although ISO has long-been reported as "safe" and to have less impact on cardiac function than other commonly-used anesthetics, recent findings showed that ISO-induced inhalational anesthesia actually could lead to a variety of cardiovascular/cardiopulmonary side effects [4, 5]. Those findings contrast with other reports of beneficial 'side effects' from host exposures to ISO [13] as well as two studies that reported ISO use could help protect the heart against ischemia-re-perfusion (I/R) injury [14, 15].

**Day 1**

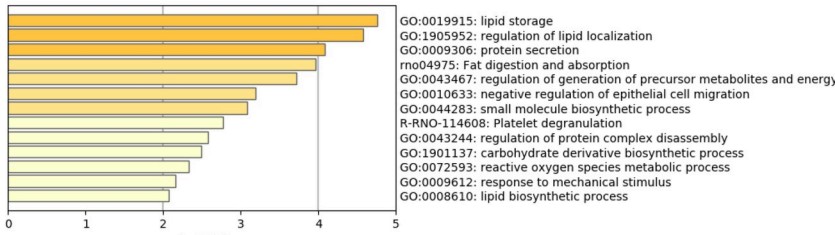

**Day 30**

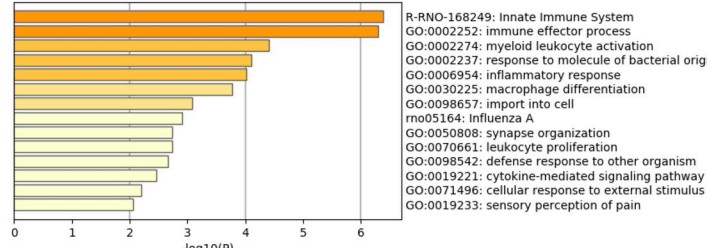

**Day 240**

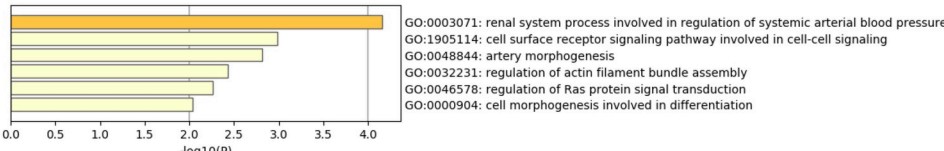

**Day 360**

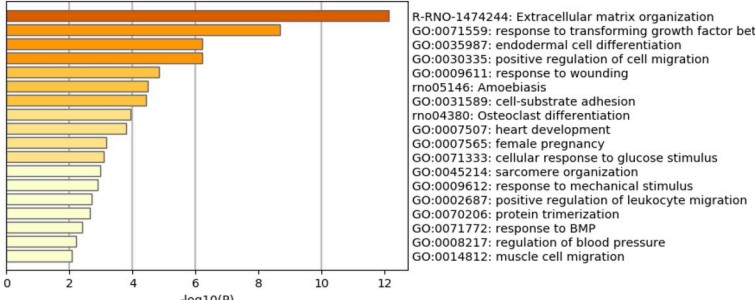

**Fig 15. Enrichment analysis for up-regulated genes in hearts.** Functional gene set enrichment analysis (GSEA) of top up-regulated genes was performed for tissues collected at Days 1, 30, 240, and 360 post-exposures. The X-axis is the negative log-transformed p-value of the enriched terms. The darker the color, the more statistically significant the enriched term (pathways, GO terms, etc.). Groups of rats exposed to air (naïve, n = 6/each timepoint group) or ISO (n = 5/Day 1 group, and n = 6/Day 30, 240, and 360 groups) were used.

ISO is known to reduce arterial blood pressure in a dose-related manner in human and also in laboratory animals. Kouki et al. (2016) reported strain-specific isoflurane-induced cardio-respiratory effects between rat strains. They showed that SHR rats are more sensitive to suppressive effects of ISO on cardiovascular function among rat strains (SHR, control strain Wistar Kyoto rats (WKY), and normotensive Sprague-Dawley (SD) rats) [40]. Therefore, DEG patterns in our data might be different in healthy animals. To date, there are only a handful of

**Day 1**

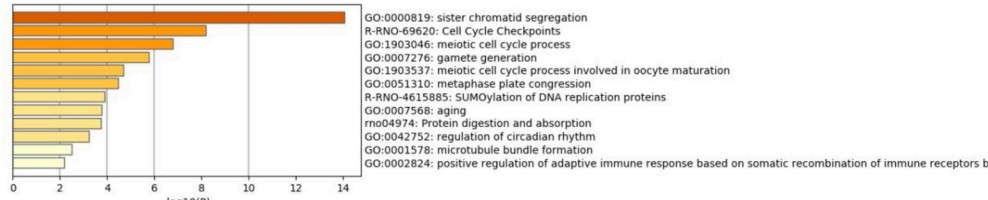

**Day 30**

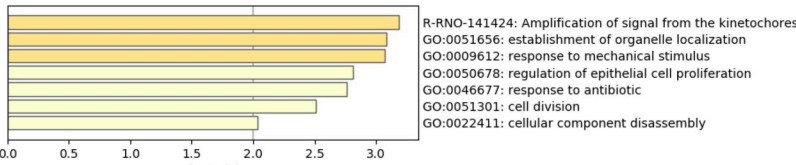

**Day 240**

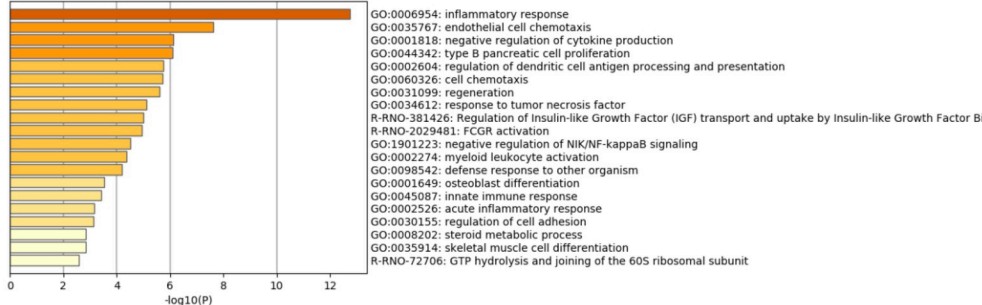

**Day 360**

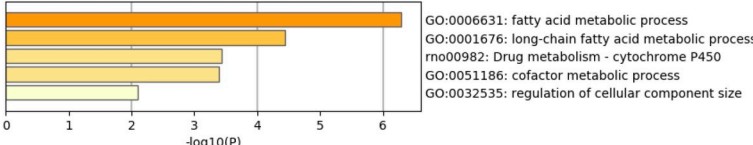

**Fig 16. Enrichment analysis for down-regulated genes in hearts.** Functional gene set enrichment analysis (GSEA) of top down-regulated genes was performed for tissues collected at Days 1, 30, 240, and 360 post-exposures. The X-axis is the negative log-transformed p-value of the enriched terms. The darker the color, the more statistically significant the enriched term (pathways, GO terms, etc.). Groups of rats exposed to air (naïve, n = 6/each timepoint group) or ISO (n = 5/Day 1 group, and n = 6/Day 30, 240, and 360 groups) were used.

reports showing which genes and pathways ISO exposures might influence in host heart tissues. Previous studies in healthy animals showed that ISO could affect host heart tissues. One report showed that ISO exposures could increase on levels of reactive oxygen species (ROS) and activate Nrf2-anti-oxidant response element signaling pathways, thereby attenuating myocardial IR injury in rats [41]. Another report showed that ISO exposures could protect cardiac function in rats against myocardial ischemia-re-perfusion injury via inhibition of p38 MAPK signaling pathways [42]. These changes ultimately led to reductions in the area of myocardial infarction, alleviated the pathological damage in myocardial cells, and reduced the oxidative

stress response in the organ [42]. Another report showed that a partial decrease in mitochondrial membrane potential ($\Delta\Psi$(m)) was an underlying basis for some of the cardioprotective effects resulting from ISO exposures [43]. It was believed that these effects arose from attenuation of excess mitochondrial ROS production and consequential decreases in mitochondrial calcium ion uptake.

Several reports have shown that ISO exposure(s) cause apoptosis in general, and impact on apoptotic pathways in particular, in a variety of cells including cardiomyocytes and endo-the-lial cells [27, 28, 44, 45]. For the most part, studies on whether/how ISO causes apoptotic damage have mostly focused on nerve cells in the brain/spinal cord, especially in those of newborn and/or developing hosts [44, 46, 47]. Nevertheless, whether ISO causes a consistent pattern of increase or decrease in this phenomenon in heart tissues or nerve cells associated with heart function remains unclear.

As noted above, the over-arching demonstrable effect from *in vivo* ISO exposure(s) has been the development of a variety of neurotoxicities in the hosts. In some cases, the documented toxicities were often dependent on ISO concentration, duration of exposure(s), and/or gender\age of the exposed host [20, 24, 48–58]. Whether such effects (with associated constraints) occur at the level of cardiac tissue cellular components, i.e., endothelial cells, myocytes, etc., remains unstudied, to date. Again, many of these ISO-induced toxicities were believed related to induction of changes in a variety of cell signaling pathways (i.e., relative to oxidative stress [20, 24, 41, 42], cell cycle [53, 59], immune function and inflammation [28, 51]). Those findings align with some of the data from the current study showing that genes for components within these pathways were also impacted in time-related manners in the ISO-exposed SHR hosts.

Several anesthetics, including ISO, have been shown to affect glucose and lipid metabolism in exposed hosts (apart from any effects on heart function) [60–64]. Several reports have shown that ISO exposure(s) caused an immediate metabolic effect and changes in fatty acid metabolic pathways. ISO exposures were also shown to cause increased blood glucose levels and decreased blood insulin levels in mice [65]. Another study showed that ISO exposures led to detrimental effects on circadian metabolism and alterations in plasma levels of glucose, lactic acid, blood gases, total fatty acids (TFA), melatonin, insulin, and corticosterone in Sprague-Dawley rats [66]. In another study using dog models, levels of synthesis of proteins, fatty acids, and glucose were decreased following ISO exposures [67]. Together, these cited findings align with some of the data obtained here showing that genes for a number of the components within these pathways were impacted in time-related manners in the ISO-exposed SHR hosts. Whether/how these induced changes in protein, carbohydrate, and fatty acid synthesis-metabolic pathways give rise to potential changes in heart function in the SHR rats is part of a series of questions being addressed in follow-up studies in our laboratories.

In conclusion, to determine if repeated (i.e., one instillation each two days) ISO exposures impart acute and/or long-lasting effects in the heart, gene expression in LV recovered from SHR hosts were analyzed at various timepoints over a 1-yr post-exposure period. Such changes could have important implications for use of this anesthetic when treating patients susceptible to heart failure or other cardiac abnormalities. These data revealed there were a variety of DEG in response to repeated acute ISO exposure at all post-exposure timepoints in this rat species. Such exposures led to DEG associated with, among other things, tissue (damage) healing, local immune function, inflammation, and circadian rhythm regulation at Days 1 and 30 post-exposure; however, these specific gene effects mostly dissipated by Day 240.

Nevertheless, at Day 360 there were still significantly increased numbers of other genes DEG induced by the repeated acute ISO exposures. Among these were several associated with cell signaling, differentiation, and migration, extracellular matrix organization, cell-substrate

adhesion, heart development, and regulation of blood pressure. In addition, analysis of consistent DEG at Days 240 and 360 post-exposure indicated the presence of late onset DEG that potentially could reflect long-term effects from the acute ISO exposures. These included several DEG associated with a variety of important pathways, including oxidative phosphorylation, ribosome, angiogenesis, mitochondrial translation elongation, and focal adhesion. The current data showed that Col4a2 and Col4a1 were found among the late onset DEG associated with the angiogenesis pathway. Several reports have shown that mutations in Col4a2 and Col4a1 are linked to heart diseases [68, 69]. Taken together, the data show that repeated acute exposures of SHR rats to ISO could impart variable effects on gene expression/regulation in the heart and while some self-resolve, others appear to be long-lasting. Whether such changes occur in all rat models or in humans remains to be investigated. In addition, detailed histological analyses in the future will allow us to better to understand what kind of tissue remodeling might be going on that would lead to these changes in gene expression or are induced by these changes.

One concern is that the 'instilled' tube itself during an intratracheal inhalation (ITIH) is somehow causative of the responses. There seem two ways to try to potentially address the concern: (1) intubate rats 2 hr/2 days while under some other anesthetic during ITIH to see if 'same' effects are seen, thereby implicating the Insyte tube, or (2) place rats in a chamber to allow for 2 hr nose-only breathing of ISO (starting at 5% and then maintained at 2.5%) to see if the tube caused ALL the observed effects here. The first approach is ruled out as there would be no way to control for anesthetic-specific effects that would hamper comparisons of results (refer to multiple papers that compare ISO vs. sevoflurane vs. desflurane inhalants) [70, 71] and so would not allow tube-specific effects to be discerned clearly. The second approach, while ultimately do-able, would basically reiterate findings from a paper by DeLano and Zeifach (1981) [72] that noted some changes in systemic blood pressure in SHR rats (vs. WKY rats) after anesthesia (not ISO) and surgical intubation (open trachea and then insert tube; not method here). However, any effects from the intubation were short-lived and decreased over the period of time the tube was present (within hours). The authors could not explain the transient pressor effect in the SHR rats and noted that any similar effect in humans was mitigated by coating the tube with lidocaine. A more recent study (Konno et al. 2014) [73] found endotracheal intubation of rats under ISO (and then maintained under ISO) for periods beyond 30 min had no impact on vital signs and caused no histopathological trauma to the lungs or the trachea.

Thus, while we might concede that some of the reported outcomes here at Day 1 (i.e., wound-healing pathways activated, inflammation) could potentially be tied back to the presence of the Insyte tube for those two 2-hr periods, those expression changes were gone by > Day 30 post-exposure and did not return. We do not concede that the very short presence of a tube in the upper trachea of each rat was what led to the outcomes seen in the heart tissues at Days 240 and 360 post-exposure.

In conclusion, the present data do not allow us to make determinations about the potential impact on cardiac genes following single (non-repeated) low-dose exposures to ISO as used in many in-patient protocols. On the other hand, repeated ISO use has been employed over time in the sedation of critically Ill children [74] and in the post-operative intensive care of some cardiac surgery patients [75]. ISO has also been seen as one of the volatile anesthetic agents for use in long-term critical care sedation (VALTS) of intensive care unit (ICU) patients [76]. To date, no data corresponding to the long-term cardiac health of these types of ISO-treated patients has been reported in the literature. It is hoped the results from this study will provide clinicians and others a cautionary note regarding any potential overuse/overly-prolonged use of ISO.

## Supporting information

**S1 Fig. PCA plots of samples using all genes at Day 1, 30, 240, 360.** Principal component analyses (PCA) were conducted on normalized log-transformed expression values from ISO and Naïve samples at each timepoint. The X-axis is the first principal component (PC), Y-axis is the second PC. The green dots represent ISO samples, and orange triangles refer to naïve rats. The ellipses were made under a 95% confidence level. The PCA plots show no evidence of outliers.
(TIFF)

**S1 Table. Top genes increased in hearts due to ISO exposure (relative to naïve) on Day 1.** Top genes significantly up-regulated between ISO and naive rats at Day 1 are listed.
(DOCX)

**S2 Table. Top genes increased in hearts due to ISO exposure (relative to naïve) on Day 30.** Top genes significantly up-regulated between ISO and naive rats at Day 30 are listed.
(DOCX)

**S3 Table. Top genes increased in hearts due to ISO exposure (relative to naïve) on Day 240.** Top genes significantly up-regulated between ISO and naive rats at Day 240 are listed.
(DOCX)

**S4 Table. Top genes increased in hearts due to ISO exposure (relative to naïve) on Day 360.** Top genes significantly up-regulated between ISO and naive rats at Day 360 are listed.
(DOCX)

**S5 Table. Top genes reduced in hearts due to ISO exposure (relative to naïve) on Day 1.** Top genes significantly down-regulated between ISO and naive rats at Day 1 are listed.
(DOCX)

**S6 Table. Top genes reduced in hearts due to ISO exposure (relative to naïve) on Day 30.** Top genes significantly down-regulated between ISO and naive rats at Day 30 are listed.
(DOCX)

**S7 Table. Top genes reduced in hearts due to ISO exposure (relative to naïve) on Day 240.** Top genes significantly down-regulated between ISO and naive rats at Day 240 are listed.
(DOCX)

**S8 Table. Top genes reduced in hearts due to ISO exposure (relative to naïve) on Day 360.** Top genes significantly down-regulated between ISO and naive rats at Day 360 are listed.
(DOCX)

## Author Contributions

**Conceptualization:** Sung-Hyun Park, Yuting Lu, Yongzhao Shao, Lung-Chi Chen, Mitchell D. Cohen.

**Data curation:** Sung-Hyun Park, Yuting Lu, Yongzhao Shao.

**Formal analysis:** Yuting Lu, Yongzhao Shao.

**Investigation:** Sung-Hyun Park, Yuting Lu, Yongzhao Shao.

**Methodology:** Sung-Hyun Park, Yuting Lu, Yongzhao Shao, Colette Prophete, Lori Horton, Maureen Sisco, Hyun-Wook Lee, Thomas Kluz, Hong Sun.

**Writing – original draft:** Sung-Hyun Park.

**Writing – review & editing:** Sung-Hyun Park, Yuting Lu, Yongzhao Shao, Hyun-Wook Lee, Hong Sun, Max Costa, Judith Zelikoff, Lung-Chi Chen, Mitchell D. Cohen.

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
