## [Decision Letter · Decision Letter 0]

3 May 2021

PONE-D-21-09743

Longitudinal Impact on Rat Cardiac Tissue Transcriptomic Profiles due to Acute Intratracheal Inhalation Exposures to Isoflurane

PLOS ONE

Dear Dr. Park,

Thank you for submitting your manuscript to PLOS ONE. After careful consideration, we feel that it has merit but does not fully meet PLOS ONE’s publication criteria as it currently stands. Therefore, we invite you to submit a revised version of the manuscript that addresses the points raised during the review process.

Two expert reviewers have studied your manuscript. Although both find the topic of interest they have made several comments that you will need to address in a revised manuscript. Particularly, reviewer #1 has identified flaws in your experimental design that can only be repaired by including additional groups. Hence, experiments should be redone in SHR rats receiving free, spontaneous inhalation with ISO compared to air only. In addition, control WKY rats receiving free, spontaneous inhalation with ISO compared to air only, should also be included. However, I find it acceptable if these additional groups are not followed for 240 and 360 days. The earlier time points (D1 and D30) will suffice.

If you will need more time than 6 months to complete your revisions, please reply to this message or contact the journal office at plosone@plos.org. Please include the following items when submitting your revised manuscript:

We look forward to receiving your revised manuscript.

Kind regards,

Jaap A. Joles, DVM, PhD

Academic Editor

PLOS ONE

Journal Requirements:

Reviewers' comments:

Reviewer's Responses to Questions

**Comments to the Author**

1. Is the manuscript technically sound, and do the data support the conclusions?

Reviewer #1: No

Reviewer #2: Partly

2. Has the statistical analysis been performed appropriately and rigorously? 

Reviewer #1: Yes

Reviewer #2: I Don't Know

3. Have the authors made all data underlying the findings in their manuscript fully available?

Reviewer #1: Yes

Reviewer #2: No

4. Is the manuscript presented in an intelligible fashion and written in standard English?

Reviewer #1: Yes

Reviewer #2: Yes

5. Review Comments to the Author

Reviewer #1: Aim of the study described in the manuscript by Park et al., is to elucidate the effect of the inhalation anesthetic isoflurane (ISO) on gene expression patterns of cardiac tissue. While ISO may not lead to long-lasting side-effects, short- and long-term effects on gene regulation remain unexplored.

Park et al., investigated the role of ISO on SHR rats by 2 consecutive days of 2hour ISO (2%) intra-trachael inhalation (IHIH) versus air only inhalation. Hearts were explanted on D1, D30, D240 and D360 post-exposure. Their main finding was that ISO exposure leads to changes in cardiac gene expression, some of which last up until D360. The primary pathways involved included general immune response e.g. wound healing and cytokine-mediated pathways, circadian rhythm, regulation of systemic arterial blood pressure, fatty acid metabolism and extracellular matrix organization.

I appreciate the effort put into this manuscript as indeed inhalation anesthetics, often used when performing experimental cardiovascular surgeries on rodents, may unintentionally influence (cardiac) cellular processes. However, in most of these cases, both controls/Shams and experimental intervention groups will receive the same (type of) anesthetic treatment. In this manuscript, the methods describe that control animals did not receive IHIH. The real comparison here is thus SHR + spontaneous breathing versus SHR + mechanical breathing on ISO. I believe it is thus not possible to discern between the effects of ISO vs IHIH on cardiac gene expression and thus impossible to attribute the findings to ISO alone.

Please find a more detailed overview of my comments below.

Major:

1) Regarding the experimental set-up: from the manuscript it becomes clear that the ISO but not the control group received IHIH. There is evidence that mechanical ventilation affects cardiovascular functioning (e.g. PMID: 2199000 and PMID: 10208959). So how sure are the authors that the induced changes in gene expression are due to ISO and not to IHIH? I think the fact that for instance the wound-healing pathways are activated, says more about the effect of introducing IHIH than the effect of ISO. Physical injury due to multiple intra-tracheal insertions could possibly explain a whole range of the DEGs and pathways found in this manuscript. In order to be really able to investigate the sole effect of ISO, experiments should be redone in SHR rats receiving free, spontaneous inhalation with ISO compared to air only.

2) There is indeed evidence that ISO inhalation results in altered cardiovascular function. Examples include concentration-dependent decreases in heart rate and blood pressure (PMID: 10622172) which may lead to more severe hypotension in hypertensive circumstances (PMID: 15574546). Others have also shown that vascular effects ex vivo and in vivo in hypertensive rats are more severe than in normotensive rats (PMID: 15574546). However, if cardiac gene expression is equally affected by ISO remains unknown. As outcomes in SHR are generally compared to e.g. WKY rats, authors should also consider taking into account the DEGs and pathways that RNA-seq yields for these animals on ISO versus control.

3) The methodological approach and pathway analyses based on the RNA-seq data looks sound.

Reviewer #2: Title: Longitudinal Impact on Rat Cardiac Tissue Transcriptomic Profiles due to Acute

Intratracheal Inhalation Exposures to Isoflurane

This is a remarkable study, with time points reaching 360 days. It is indeed uncommon to see studies go into that level of depth with their measurements. It is yet more remarkable that there is any effect at this time point. This study is very intriguing and will undoubtedly be impactful in the field, and I congratulate the authors on this monumental effort. However, I do think that several things need to be added to this paper to show individual gene measurements and plots of variance – for example showing some replicates for the later time points would reassure readers that these time points are real measurements and provide more convincing evidence of these striking findings.

6. PLOS authors have the option to publish the peer review history of their article (what does this mean?). If published, this will include your full peer review and any attached files.

Reviewer #1: No

Reviewer #2: No

---

## [Author Response · Author response to Decision Letter 0]

25 Jun 2021

The Authors are very grateful to the Reviewers for their comments. We have reviewed all the comments in preparing this revised version. The responses to reviewers and editor’s comments are addressed in a point-by-point style: our responses in the 'Response to Reviewers' file are indicated in blue. Changes that have been made in the Revised Manuscript are indicated in yellow highlight.

---

## [Decision Letter · Decision Letter 1]

13 Jul 2021

PONE-D-21-09743R1

Longitudinal Impact on Rat Cardiac Tissue Transcriptomic Profiles due to Acute Intratracheal Inhalation Exposures to Isoflurane

PLOS ONE

Dear Dr. Park,

Thank you for submitting your manuscript to PLOS ONE. After careful consideration, we feel that it has merit but does not fully meet PLOS ONE’s publication criteria as it currently stands. Therefore, we invite you to submit a revised version of the manuscript that addresses the points raised during the review process.

Both reviewers find the revised manuscript improved. Note that reviewer #2 has submitted a separate file with comments.

Nevertheless, both still have some remaining comments that you must address to allow acceptance of this manuscript. 

We look forward to receiving your revised manuscript.

Kind regards,

Jaap A. Joles, DVM, PhD

Academic Editor

PLOS ONE

Reviewers' comments:

Reviewer's Responses to Questions

**Comments to the Author**

1. If the authors have adequately addressed your comments raised in a previous round of review and you feel that this manuscript is now acceptable for publication, you may indicate that here to bypass the “Comments to the Author” section, enter your conflict of interest statement in the “Confidential to Editor” section, and submit your "Accept" recommendation.

Reviewer #1: (No Response)

Reviewer #2: All comments have been addressed

2. Is the manuscript technically sound, and do the data support the conclusions?

Reviewer #1: Partly

Reviewer #2: Yes

3. Has the statistical analysis been performed appropriately and rigorously? 

Reviewer #1: Yes

Reviewer #2: Yes

4. Have the authors made all data underlying the findings in their manuscript fully available?

Reviewer #1: Yes

Reviewer #2: No

5. Is the manuscript presented in an intelligible fashion and written in standard English?

Reviewer #1: Yes

Reviewer #2: Yes

6. Review Comments to the Author

Reviewer #1: Aim of the study described in the manuscript by Park et al., is to elucidate the effect of the inhalation anesthetic isoflurane (ISO) on gene expression patterns of cardiac tissue. Short- and long-term effects on gene regulation by ISO, especially in models of cardiovascular disease, remain largely unexplored.

Park et al., investigated the role of ISO on SHR rats by 2 consecutive days of 2hour ISO (2%) intra-trachael inhalation (IHIH) versus air only inhalation. Hearts were explanted on D1, D30, D240 and D360 post-exposure. Their main finding was that ISO exposure leads to changes in cardiac gene expression. In their revised manuscript, the author place more emphasis on acute versus (novel) long-term effects, hereby discerning the differential pathways induced by ISO exposure.

I appreciate the additional effort put in the revised manuscript, both in writing and in data. The additional heatmaps, Volcano-plots and GO-pathway analysis give a more in-depth overview of the RNA-seq data.

Please find a more detailed overview of my comments below.

1) Regarding the experimental set-up: the authors have shared their thoughts and considerations on choosing this specific approach (eg. IHIH vs free-breathing). I would suggest to include part of this information in the Materials and Methods sections and discussion. This can be done to both emphasize and discuss the differences in acute (eg <d30) and="" long-term="">D240) DEG patterns.

2) In general, the authors have made a clearer distinction in representing and discussing the acute and long-term DEG patterns. This gives the reader a better insight between the types of pathways regulated. Indeed, I believe the long-term, newly manifested, effects are the most interesting; regulation of angiogenesis, extracellular matrix organization, (OXPHOS and FA) metabolism and cell signaling. While I understand that the authors will dedicate a separate manuscript on how these changes may lead to heart function alterations, lacks information on how the SHR model relates to previous work in healthy animals, which links to point 3) and the question of adding a WKY control. If the authors primary goal is to share information on routine ISO employment in a variety of clinical settings, patients at risk should be identified. Now all previous models and spatio-temporal changes are equally discussed.

3) The discussion lacks connections between existing literature and the current study on potential mechanisms of ISO to induce these DEG patterns. Specifically by now separating the acute and long-term phase, some explanation should be included how these transient changes come into play.

4) Novel text addition lines 531 – 536 seems to have overlapping message with lines 537 – 541. The distinction between these statements is better described in line 604 – 611.)

Reviewer #2: This is an interesting work. Overall I support the publication of these data as an important addition to the scientific record. Please see my attached detailed comments for minor revisions to improve the communication of these findings to the reader and clarify several issues I feel may be important for the reader.

7. PLOS authors have the option to publish the peer review history of their article (what does this mean?). If published, this will include your full peer review and any attached files.

Reviewer #1: No

Reviewer #2: **Yes: **Matthew R. Sapio

---

## [Author Response · Author response to Decision Letter 1]

5 Aug 2021

Thank you very much for considering our MS “Longitudinal Impact on Rat Cardiac Tissue Transcript-omic Profiles due to Acute Intratracheal Inhalation Exposures to Isoflurane”. The Authors are very grateful to the Reviewers for their comments. We have reviewed all the comments in preparing this revised version. The responses to reviewers’ comments are addressed in a point-by-point style: our responses are indicated in blue. Changes that have been made in the Revised Manuscript are indicated in yellow highlight.

We hope this revision has addressed all the comments and that the revised MS is now acceptable for publication in PLOS One. If there still any questions, please do not hesitate to contact me.

---

## [Decision Letter · Decision Letter 2]

18 Aug 2021

PONE-D-21-09743R2

Longitudinal Impact on Rat Cardiac Tissue Transcriptomic Profiles due to Acute Intratracheal Inhalation Exposures to Isoflurane

PLOS ONE

Dear Dr. Park,

Thank you for submitting your manuscript to PLOS ONE. After careful consideration, we feel that it has merit but does not fully meet PLOS ONE’s publication criteria as it currently stands. Therefore, we invite you to submit a revised version of the manuscript that addresses the points raised during the review process.

Reviewer #1 has a few additional remarks that should not form a problem.

We look forward to receiving your revised manuscript.

Kind regards,

Jaap A. Joles, DVM, PhD

Academic Editor

PLOS ONE

Journal Requirements:

Additional Editor Comments (if provided):

Reviewers' comments:

Reviewer's Responses to Questions

**Comments to the Author**

1. If the authors have adequately addressed your comments raised in a previous round of review and you feel that this manuscript is now acceptable for publication, you may indicate that here to bypass the “Comments to the Author” section, enter your conflict of interest statement in the “Confidential to Editor” section, and submit your "Accept" recommendation.

Reviewer #1: All comments have been addressed

Reviewer #2: All comments have been addressed

2. Is the manuscript technically sound, and do the data support the conclusions?

Reviewer #1: Yes

Reviewer #2: Yes

3. Has the statistical analysis been performed appropriately and rigorously? 

Reviewer #1: Yes

Reviewer #2: Yes

4. Have the authors made all data underlying the findings in their manuscript fully available?

Reviewer #1: Yes

Reviewer #2: Yes

5. Is the manuscript presented in an intelligible fashion and written in standard English?

Reviewer #1: No

Reviewer #2: Yes

6. Review Comments to the Author

Reviewer #1: Aim of the study described in the manuscript by Park et al., is to elucidate the effect of the inhalation anesthetic isoflurane (ISO) on gene expression patterns of cardiac tissue. Short- and long-term effects on gene regulation by ISO, especially in models of cardiovascular disease, remain largely unexplored.

The additional changes from this second round have further improved the content and message of the manuscript. My comments therefore primarily concern writing to improve clarity and readability.

1) Line 122 - 125 repetition of materials and methods with lines 136 – 137 but final concentration of ISO in not consistent (2% vs 2,5%)

2) It is common that the total amount of animals, eg animals per timepoint per group are described in the materials and methods (line 114 onwards) and not just the figure legends. I assume the Figure legend 1A means n=6 or n=5-6 animals per group PER TIMEPOINT. Ideally, use of animals is repeated for each Figure.

3) Materials and methods from line 126: text explaining the use of ITIH because of the WTC-dust experiments confuses me. Do the authors mean that SHR animals on ITIH were part of a control for WTC-dust trials? I think it must be explicitly mentioned that the animals in this manuscript were not exposed to WTC-dust themselves.

4) Caption Table 6, Line 463; addition of ‘Late onset’ DEG

5) Line 562 – 553: ‘Kouki et al. (2016) reported strain-specific isoflurane (…)’

6) Line 639 – 640: missing references about comparisons to other inhalant anesthetics

7) Line 654: Copy-paste of response to reviewer. I would remove ‘respectfully’

8) Line 659: Copy-paste of response to reviewer. I would remove ‘ -and still do-‘ and change to ‘Thus we believe’

9) Insertion of lines 657 – 664 largely overlapping in content with following lines 667 – 673.

Reviewer #2: I have no further revisions. I recommend the publication of this manuscript if there are no other comments from other reviewers or the editor.

7. PLOS authors have the option to publish the peer review history of their article (what does this mean?). If published, this will include your full peer review and any attached files.

Reviewer #1: No

Reviewer #2: **Yes: **Matthew R. Sapio

---

## [Author Response · Author response to Decision Letter 2]

24 Aug 2021

Reviewer #1

Aim of the study described in the manuscript by Park et al., is to elucidate the effect of the inhalation anesthetic isoflurane (ISO) on gene expression patterns of cardiac tissue. Short- and long-term effects on gene regulation by ISO, especially in models of cardiovascular disease, remain largely unexplored.

The additional changes from this second round have further improved the content and message of the manuscript. My comments therefore primarily concern writing to improve clarity and readability.

1) Line 122 - 125 repetition of materials and methods with lines 136-137 but final concentration of ISO in not consistent (2% vs 2.5%) Response: We thank the Reviewer for the comment. The concentration was corrected to 2.5% in the entire manuscript.

2) It is common that the total amount of animals, e.g., animals per timepoint per group, are described in the Materials and Methods (line 114 onwards) and not just the figure legends. I assume the Figure legend 1A means n = 6 or n = 5-6 animals per group PER TIMEPOINT. Ideally, use of animals is repeated for each Figure. Response: The animal numbers used has been added in the Materials and Methods (line 114), as well as in each figure legend. These values are per group per timepoint.

3) Materials and methods from line 126: text explaining the use of ITIH because of the WTC-dust experiments confuses me. Do the Authors mean that SHR animals on ITIH were part of a control for WTC-dust trials? I think it must be explicitly mentioned that the animals in this manuscript were not exposed to WTC-dust themselves. Response: This clarifying text has been added.

4) Caption Table 6, Line 463; addition of ‘Late onset’ DEG Response: The text was added.

5) Line 562-553: ‘Kouki et al. (2016) reported strain-specific isoflurane (…)’ Response: The text was modified.

6) Line 639-640: missing references about comparisons to other inhalant anesthetics Response: The missing references have been added.

7) Line 654: Copy-paste of response to reviewer. I would remove ‘respectfully’ Response: The text was modified.

8) Line 659: Copy-paste of response to reviewer. I would remove ‘ -and still do-‘ and change to ‘Thus we believe’ Response: As lines 657-664 were deleted (see comment #9), the text was deleted.

9) Insertion of lines 657-664 largely overlapping in content with following lines 667-673. Response: See above.

Reviewer #2

I have no further revisions. I recommend the publication of this manuscript if there are no other comments from other reviewers or the editor. Response: Thank you.

---

## [Editor Report · Decision Letter 3]

27 Aug 2021

Longitudinal Impact on Rat Cardiac Tissue Transcriptomic Profiles due to Acute Intratracheal Inhalation Exposures to Isoflurane

PONE-D-21-09743R3

Dear Dr. Park,

We’re pleased to inform you that your manuscript has been judged scientifically suitable for publication and will be formally accepted for publication once it meets all outstanding technical requirements.

Kind regards,

Jaap A. Joles, DVM, PhD

Academic Editor

PLOS ONE
---

## [Editor Report · Acceptance letter]

15 Sep 2021

PONE-D-21-09743R3 

Longitudinal Impact on Rat Cardiac Tissue Transcriptomic Profiles due to Acute Intratracheal Inhalation Exposures to Isoflurane 

Dear Dr. Park:

I'm pleased to inform you that your manuscript has been deemed suitable for publication in PLOS ONE. Congratulations! Your manuscript is now with our production department. 

Kind regards, 

on behalf of

Dr. Jaap A. Joles 

Academic Editor

PLOS ONE